



# Does the coupling of the mesospheric semiannual oscillation with the quasi-biennial oscillation provide predictability of Antarctic sudden stratospheric warmings?

Viktoria J. Nordström[1] and Annika Seppälä[1]

[1]Department of Physics, University of Otago, Dunedin, New Zealand

**Correspondence:** Annika Seppälä (annika.seppala@otago.ac.nz)

**Abstract.** During September 2019 there was a sudden stratospheric warming over Antarctica, which brought disruption to the usually stable winter vortex. The mesospheric winds reversed and temperatures in the stratosphere rose by over 50 K. Whilst this was only the second SSW in the Southern Hemisphere (SH), the other having occurred in 2002, its Northern counterpart experiences about six per decade. Currently, an amplification of atmospheric waves during winter is thought to trigger SSWs.

Our understanding, however, remains incomplete, especially with regards to its occurrence in the SH. Here, we investigate the interaction of two equatorial atmospheric modes, the Quasi Biennial Oscillation (QBO) and the Semiannual Oscillation (SAO) during the SH winters of 2019 and 2002. Using MERRA-2 reanalysis data we find that the two modes interact at low latitudes during their easterly phases in the early winter, forming a zero wind line that stretches from the lower stratosphere into the mesosphere. This influences the meridional wave guide, resulting in easterly momentum being deposited in the mesosphere

throughout the polar winter, reducing the magnitude of the westerly winds. As the winter progresses these features descend into the stratosphere, until SSW conditions are reached. We find similar behaviour in two other years leading to delayed dynamical disruptions later in the spring. The timing and magnitude of the SAO and the extent of the upper stratospheric easterly QBO signal, that results in the SAO-QBO interaction, was found to be unique in these years, when compared to the years with a similar QBO phase. We propose that this early winter behaviour may be a key physical process in decelerating

the mesospheric winds which may precondition the Southern atmosphere for a SSW. Thus the early winter equatorial upper stratosphere-mesosphere together with the polar mesosphere may provide critical early clues to an imminent SH SSW.

## 1 Introduction

During the austral winter of 2019, the Southern Hemisphere (SH) experienced a rare atmospheric disruption known as a sudden stratospheric warming (SSW) (for a recent comprehensive review on SSWs see Baldwin et al., 2021). Between September 5-11

temperatures in the Antarctic stratosphere warmed by 50 K (Yamazaki et al., 2020). Furthermore, the polar mesospheric winds reversed, and the easterlies reached ∼60 m/s around September 16 (Rao et al., 2020). The SSW's aftermath cascaded down through the atmosphere for months following its occurrence. The 2019 SSW pushed the Southern Annual Mode (SAM) into a negative phase (Rao et al., 2020), which signifies a shift of polar westerlies towards the equator (Doddridge and Marshall, 2017). This movement of strong westerly winds is believed to have impacted the Australian wildfires, which began in the





following November (Lim et al., 2019). Furthermore, the changes in polar temperatures and winds shrunk the ozone hole to
its smallest size ever observed (Eswaraiah et al., 2020a). Most of our understanding about SSWs comes from their occurrence
over the Arctic, where they take place almost every other year (Charlton and Polvani, 2007). This latest SH SSW is only one
of two that has occurred since record keeping began (Rao et al., 2020), and provides a unique opportunity to investigate the
triggers of SH SSWs.

As winter descends on the pole, the increase in the meridional temperature gradient accelerates the westerly winds into a
vortex (Brasseur, 2005). This isolates the polar air, trapping ozone destroying chemicals, causing the ozone hole over Antarctica
(Solomon, 1999), and more recently during strong Northern Hemisphere vortex events, also over the Arctic (Manney et al.,
2020; Wohltmann et al., 2020). The vortex dissipates in the spring as the atmospheric temperatures warm, in an event known as
the final warming (Schoeberl and Newman, 2014). However, during a sudden stratospheric warming the vortex breaks up earlier

than expected. Within days of the SSW's onset the vortex weakens, becomes contorted and can break up (Holton, 2012). There
is a large-scale warming of the polar stratosphere, which can result in a reversal of the meridional temperature gradient, creating
an easterly zonal wind (Volland, 1988). The World Meteorological Organization (WMO) separates SSWs into two types, major
and minor (Charlton and Polvani, 2007). It should be noted that these definitions are based on Northern Hemisphere (NH)
SSW events (Butler et al., 2015). Major events occur when the mean zonal winds at 60° latitude and 10 hPa reverse, and the

stratospheric temperature increases, often between 30-50 K (Volland, 1988). Minor events have similar temperature changes to
major warmings, however, there is no wind reversal at 60° latitude and 10 hPa (Volland, 1988). Another classification separates
SSWs based on their contortion of the polar vortex (Charlton and Polvani, 2007). A "vortex displacement" shifts the vortex off
the pole, with the vortex forming "comma shape". A "vortex split" sees the vortex break up into two pieces of comparable size.

The impacts of SSWs can influence the polar atmosphere for months (see e.g. Baldwin and Dunkerton, 2001). For ex-
ample, SSWs contribute to the size of the ozone hole via two different mechanisms. First, the warming of the stratosphere
suppresses the formation of polar stratospheric clouds (Shen et al., 2020). These clouds are where ozone depleting reactions
occur (Solomon, 1999). Furthermore, the weakening of the vortex allows the intrusion of ozone rich mid-latitude air into the
pole. Both these effects in combination lead to a smaller ozone hole (Solomon et al., 1986). The SSW also impacts the Southern
and Northern Annular Modes (SAM and NAM). If easterlies appeared they can descend down into the troposphere, and push

the SAM or NAM into a negative index for months (Shen et al., 2020; Taguchi and Hartmann, 2005).

The first observed Antarctic SSW occurred in September 2002. Here, the vortex shifted off the pole and eventually split
into two. Later, one piece reformed into a weakened polar vortex (Ricaud et al., 2005). This impacted the ozone hole, which
experienced 20% less ozone loss compared to previous years (Hoppel et al., 2003). In September 2019 a SSW occurred again.
Within days, temperatures in the stratosphere increased by 50 K (Yamazaki et al., 2020). The vortex also shifted off the pole at

higher altitudes, but remained centred over the pole, albeit with weaker winds, at lower altitudes, as we will show here. Whilst
the 2002 SSW was classified as major, according to the WMO definition, the 2019 event was minor (Yamazaki et al., 2020).
Due to their rarity, the causes of a SSW in the SH are not well understood. Eswaraiah et al. (2016, 2020a, b) have further
reported of a less well known, minor SH warming in September 2010. Whilst the 2019 and 2010 are both deemed minor, their
dynamics were very different. The 2010 event had a reversal of the temperature gradient poleward of 60°S from September 15,



and the temperature increased by about 30 K at 80°S and 10 hPa (Eswaraiah et al., 2018). The zonal winds at 60°S at 10 hPa weakened by only 20-25 m/s (Eswaraiah et al., 2016). Hence it appears that the 2010 dynamics were not very similar to 2002 and 2019, which both experienced rapid warming and wind reversals.

It is widely accepted that SSWs are the product of an interaction between planetary waves and the atmospheric mean flow (Matsuno, 1971). The Arctic experiences more SSWs due to its geography. The North Pole is ringed by mountain ranges,
perfect for producing atmospheric waves (Duck et al., 2001). An enhancement of wave activity over winter causes disruption to the vortex, as the waves deposit their momentum at higher altitudes (Brasseur, 2005). However, Antarctica is enclosed by flat oceans, which don't excite waves as effectively as mountains (Holton, 2012).

As discussed by the recent comprehensive review of Baldwin et al. (2021, and references therein), the occurrence of SSW may be linked to various large scale oscillation modes in the atmosphere, including the Quasi-Biennial Oscillation (QBO,
see e.g. Anstey and Shepherd (2014)), the Semiannual Oscillation (SAO), the El Ninõ-Southern Oscillation (ENSO, see e.g. Domeisen et al. (2019b)), and the Madden Julian Oscillation (MJO, see e.g. Wheeler and Hendon (2004); Schwartz and Garfinkel (2017)). Here, we will focus on interactions between the QBO and SAO in the SH context and will not consider the ENSO and MJO in detail.

The QBO is manifested in the reversal of zonal winds in the equatorial stratosphere. The east and westward winds alternate
every 22-34 months, with an average period of 28 months (Baldwin et al., 2001). This oscillation dominates the variability of the equatorial stratosphere, however, its influence stretches to both poles (Baldwin et al., 2001). The phases of the QBO have been found to influence the polar vortex and occurrence of NH SSWs. Holton and Tan (1980) were the first to propose that the QBO modulates the subtropical zero wind line, which influences the propagation on waves in the stratosphere – a phenomenon known as the Holton-Tan effect (see Watson and Gray, 2014, and references therein). It was later discovered (concerning the
NH) that the easterly QBO phase coincides with more SSWs (Richter et al., 2011). Other known nonlinear interactions with the QBO and SSW occurrence in the NH include those with the solar cycle (Labitzke, 2005). Due to the scarcity of SH SSW events, similar relationships connecting the solar cycle, QBO and SSW occurrence have not been identified.

The Semiannual Oscillation is a switching of zonal winds in the equatorial mesosphere (Brasseur, 2005). These winds swap between westerly and easterly, with a complete cycle taking six months. These wind shears descend down from above the
mesopause into the upper stratosphere (Kawatani et al., 2020). The SAO amplitude has two peaks: one near the stratopause (1 hPa) and another close to the mesopause (0.01 hPa) (Kawatani et al., 2020). Westerlies maximise close to the equinoxes, whilst the easterlies maximise near the solstices (Brasseur, 2005). The SAO maxima at 1 hPa exhibit a seasonal asymmetry, where the 'first cycle', which begins in December with the NH easterly phase, is stronger than the 'second cycle', which starts with the SH easterly, roughly June (Garcia et al., 1997; Peña-Ortiz et al., 2010). This behaviour arises from differences in extra
tropical wave forcing, which is generally understood to be stronger in the NH winter (Garcia et al., 1997). The drivers of the SAO are not well understood. The prevailing theories suggest that the westerly accelerations, in March and September, are caused by Kelvin and high frequency gravity waves (Brasseur, 2005), whilst the easterlies maximise, during December and June, from advection of easterly momentum across the Equator, by the upper branch of the Brewer-Dobson circulation (Smith et al., 2020).





Recent work by Gray et al. (2020) noted the importance of the equatorial mesosphere and upper stratosphere on forecasting Northern Hemisphere SSWs. Their modelling study found that SSWs were only reproduced well when the flow in the equatorial upper stratosphere was constrained, simulating the two atmospheric modes in this region, the SAO and the QBO. Similar results were previously presented by Pascoe et al. (2006): In a troposphere-stratosphere-mesosphere global circulation model with forced QBO and SAO like variability, the timing of the NH mid winter warming advanced by about one month.

Whilst many studies have investigated the troposphere for answers to the questions raised by SSWs, we are now beginning to see suggestions that we should also be looking at the upper atmosphere to understand the drivers of SSWs. The works of Pascoe et al. (2006) and Gray et al. (2020) discussed above, draws attention to the upper atmosphere in the formation of a SSW, with a focus on the NH.

    Here, we report on the interaction of the QBO and SAO in the Southern Hemisphere during the winters of 2002 and 2019

(the two years with SH SSWs), based on reanalysis data. Both years exhibited clear easterly QBO conditions during the polar winter, leading to a comparison with other easterly QBO years in the SH. We find an interaction between the QBO and SAO during the austral winters of 2019 and 2002 that is unique in its timing and extent. Coinciding with the SAO-QBO interaction is an intensification of atmospheric wave propagation, which deposit easterly momentum in the upper atmosphere throughout the two winters, leading to a disturbed SH polar stratosphere and the observed SSW events.

## 2    Data and Methods

### 2.1    MERRA-2

The second Modern-Era Retrospective analysis for Research and Applications (MERRA Version 2, MERRA-2) is a National Aeronautics and Space Administration (NASA) atmospheric reanalysis product that begins in 1980 (Bosilovich et al., 2016). MERRA-2 has a horizontal resolution of $0.5° × 0.625°$ with 42 vertical levels from the surface to 0.01 hPa (Gelaro et al.,

2017). To investigate the connections between the SAO, QBO and SSW we used the four-times-daily zonal wind, geopotential height and temperature information of MERRA-2, averaged into daily means. We focus on the vertical pressure range of 550 to 0.1 hPa and the austral winter (June-July-August-September-October, JJASO).

    MERRA-2 was run in four production *Streams* (Bosilovich et al., 2016). The first three covered the periods 1980-1991 (stream 100), 1992-2000 (stream 200) and 2001-2010 (stream 300), and the final stream from 2011-present (stream 400). Each

stream had initial conditions derived from MERRA with a subsequent single year spin-up period, details of the process can be found in Bosilovich et al. (2016); Bosilovich and Coauthors (2015); Gelaro et al. (2017).

    In later analysis, were results from several years are averaged, the averaging is based on the streams. All years presented here were also analysed individually. The stream analysis ensures that decadal variability of the SAO and QBO interactions are not lost in a large average average, as the initial conditions change across the streams in MERRA2.



## 2.2 Semiannual Oscillation

The SAO is known to have a period of six months, but it has appreciable inter-annual variability (Smith et al., 2020). Here, we focus our investigation on the easterly SAO maxima that occurs close to 1 hPa during the Southern Hemisphere winter. At 1 hPa MERRA-2 has been found to represent the easterly SAO in qualitative agreement with satellite derived winds (Kawatani et al., 2020), giving confidence that the SAO representation is reasonably realistic, particularly for the changes from westerly to easterly phases, and their propagation.

## 2.3 Quasi-biennial Oscillation

To analyse SAO and QBO interactions in the upper stratosphere, we focus on years with easterly QBO (eQBO) phase specifically in the equatorial upper stratosphere in the MERRA-2 zonal mean zonal wind. Analogous to Rao et al. (2020), we take the QBO phase at the 10 hPa pressure level, which Rao et al. have shown to provide predictability in the SH SSW cases. QBO structure and dynamics in MERRA-2 reanalysis is discussed in detail by Coy et al. (2016), who conclude that MERRA-2 displays a realistic QBO behaviour in zonal winds. We verified this by contrasting to sonde observations of zonal wind from Singapore and found the two to be consistent, as expected (Coy et al., 2016).

For this study, eQBO is taken to be present if the 10 hPa equatorial flow is easterly during the austral winter months of June/July. Both years 2019 and 2002 have an upper stratospheric eQBO present during the winter. This is not the case for the aforementioned year 2010, when the equatorial upper stratosphere exhibits a westerly QBO phase.

To contrast the two SSW years to others with similar large scale equatorial flow conditions, other years with equivalent, i.e. eQBO phase, conditions during the austral winter months were analysed. The eQBO years in the MERRA-2 period were 1980, 1983, 1988, 1990, 1993, 1995, 1997, 1999, 2004, 2006, 2008, 2011, 2014, and 2017. In later analysis, the easterly QBO years have been split by model stream: i.e. 1980, 1983, 1985 and 1990 are stream 100; 1993, 1995, 1997 and 1999 are stream 200; 2004, 2006 and 2008 are stream 300; and finally, 2011, 2014 are stream 400.

We note that all years were analysed individually as well as in groups based on streams.

The years 1988 and 2017 are left out of these groups as their dynamics were found to be unique, all experienced mesospheric wind reversals in October. These years were thus analysed individually, and will be discussed separately from the other eQBO years. In 2017 the polar vortex has been reported to have experienced a disruption due to enhanced planetary wave activity throughout winter (Klekociuk et al., 2020). This lead to a smaller than average spring ozone hole (Klekociuk et al., 2020). There have also been reports of an SSW occurrence in 1988 (Kanzawa and Kawaguchi, 1990), but to our knowledge this has not been verified subsequently.

## 2.4 Wave propagation

We calculate the Eliassen-Palm flux (EP flux) from MERRA2 fields to visualise wave propagation and momentum deposit. The EP flux is a vector in the meridional plane and its direction and magnitude portray the relative importance of the eddy heat



flux and the momentum flux (Brasseur, 2005). As planetary scale waves can only propagate where the zonal flow is westerly (eastward), the location of the zero-wind line (0 m/s) forms a barrier for planetary scale wave propagation.

The upward ($F_y$) and meridional ($F_\phi$) components of EP flux are:

$$F_y = f * a * d\theta/dp * \overline{u'v'} \cos(\theta) \tag{1}$$

160

$$F_\phi = -a * \overline{v'\theta'} \cos^2(\theta) \tag{2}$$

where $f$ is the Coriolis parameter, $d\theta/dp$ is the change of potential temperature $\theta$ with respect to pressure $p$. $u$ and $v$ are the zonal and meridional winds, respectively and $a$ is the radius of the Earth. Overbar denotes a mean and $'$ indicates deviation from the mean of the parameter in question.

165 The divergence of the EP flux indicates when and where momentum is being deposited. The convergence (negative values) and divergence (positive values) of the EP flux correspond the deceleration and acceleration of zonal westerly winds (Holton, 2012). The upcoming EP flux results were calculated from the MERRA-2 data fields: temperature, eastward wind, and northward wind ($T, u, v$), according to Edmon et al. (1980), with the additional scaling for display purposes as described by Bracegirdle (2011).

## 170  3   Results

### 3.1   Minor warming of 2019

Figures 1 and 2 show the different behaviour of the polar vortex in the upper stratosphere (2 hPa, Figure 1) and the middle stratosphere (40 hPa, Figure 2), as seen in the geopotential height of the two pressure levels. During the austral winter months, June to August, the polar vortex remains over Antarctica. However, following the onset of the SSW in September, the upper 175 stratospheric vortex becomes offset from the pole, noted by the geopotential height minimum over the 120°W-40°E longitudinal sector, signifying a vortex displacement (Charlton and Polvani, 2007). At the same time, lower in the stratosphere, the minimum remains above Antarctica, as seen in Figure 2d.

Figure 3 shows how the 7 day averaged zonal mean zonal wind (coloured, filled contours) evolves with time, along with the propagation of planetary scale waves during the austral winter of 2019, with the EP flux arrows illustrating the direction of 180 wave movement. The location of the zero-wind line (contour of 0 m/s zonal mean zonal wind velocity), which forms a barrier for planetary wave propagation, is indicated as a thick white line. Furthermore, the EP flux convergence (dashed line contour) indicates where the waves dissipate and deposit easterly momentum to atmospheric flow, acting to decelerate it. The austral winter of 2019 saw the QBO in the easterly descending phase at 10 hPa (Figure 3a). Note here that the eQBO wind signal already exists around 10 hPa, and is not initiated by the descending easterly SAO (seen here above 1 hPa), as may happen 185 with wQBO (Kuai et al., 2009). During late June the easterly phase of the SAO is present in the mesosphere (Figure 3a). Poleward of the SAO signal, in the SH mesosphere, easterly momentum deposit is taking place, decelerating mesospheric flow.



Later, between July 13-19 the SAO and QBO intersect south of the equator, forming a long zero wind line at roughly 30°S that extends from 40 hPa to ~0.5 hPa (Figure 3b). Easterly momentum continues to be deposited in the mesosphere, further decelerating the zonal wind. By August 24-30 the zero wind line formed by the QBO and SAO subsides and comes to sit
between 40 and 3 hPa (Figure 3c). The zonal mean zonal wind in the mesosphere has decelerated from about 80 m/s to roughly 40 m/s, as a result of the continued momentum deposit. Furthermore, easterly momentum now continues to be deposited below and above the stratopause. Between September 7-13, when the SSW is observed in the stratosphere, zonal mean zonal wind in the mesosphere has reversed, with this easterly wind band connecting all the way through to the equatorial QBO (Figure 3d). Easterly momentum continues to be deposited around the stratopause at high polar latitudes, moving further down into the
stratosphere. Note that in this zonal mean picture the winds do not reverse down further than the mesosphere, but the winds in the stratosphere do weaken, from about 80 m/s to ~40 m/s. As will be discussed next, in this zonal average view with a longitudinal asymmetry as indicated by Figure 1, the behaviour of the EP flux arrows relative to the reversed zonal flow looks un-physical, i.e. it appears that wave propagation form the SH polar region towards the equator continues although the zonal mean zonal wind has turned easterly.

When Figure 3d is averaged in accordance to how the vortex shifted, as seen in Figure 1, clockwise from 40°E to 140°W and from 140°W to 40°E, respectively, we find more consistent behaviour in the mesosphere and upper stratosphere. This is shown in Figure 4a, where the mesospheric and upper stratospheric winds (down to about 20 hPa) at 60°S reverse on the mainly Eastern side of Antarctica, connecting into the equatorial QBO. On the largely Western side on the Antarctica, in the sector encompassing the shifted vortex, the westerlies shift towards South America, and reverse over the pole as seen in Figure 4b,
providing a pathway for wave propagation towards the equator.

An alternate view of the coupling of the SAO and QBO wind signals averaged over the latitudinal range of 15°S-20°S is shown in Figure 5, which presents how the daily zonal mean zonal winds from 100-0.1 hPa evolve over June and July. The SAO is visible in the mesosphere in June and we can see the SAO pattern descending, before the it connects to the eQBO wind pattern in July.

**3.2   Major warming of 2002**

Figure 6 is analogous to Figure 3 but now for the austral winter of 2002. Again the QBO is easterly at 10 hPa, similar to 2019. Between June 8-14 an easterly oscillation of the SAO is present in the mesosphere as seen in Figure 6a. Easterly momentum is also deposited throughout the mesosphere and upper stratosphere at roughly 60°S, leading to deceleration of the zonal winds in this region throughout the time period. In Figure 6b (June 15-21) the SAO and eQBO wind patterns intersect, similar to 2019,
well before the SSW event. The resulting zero wind line extends from roughly 30 hPa to 0.3 hPa close to 30°S, extending the barrier for planetary wave propagation into the SH mesosphere and upper stratosphere. During this time easterly momentum continues to be deposited in the mesosphere.

After this intersection, eQBO comes to lay between 50-2 hPa extending to about 40°S from the equator (Figure 6c), pushing the zero wind line with it and thus blocking wave propagation towards the equator throughout most of the upper stratosphere.
Easterly momentum continues to be deposited in the mesosphere and upper stratosphere. By this time the winds throughout





the polar atmosphere have decelerated, in the lead up to the SSW. In Figure 6d, we see the polar winds dramatically reverse between September 21-27 from 0.1 hPa down to about 50 hPa, coinciding with the 2002 stratospheric warming.

The time evolution of the zonal mean winds at 15°S-20°S during June-July 2002 is shown in Figure 7. Here we see the eQBO and easterly SAO wind patterns interacting somewhat earlier that for 2019 (Figure 5), now starting in the first half of

June.

### 3.3 Comparison to other eQBO years

With the indication that the SAO and the eQBO winds interacted in the months leading up to the SSW events in 2002 and 2019, we now proceed to investigate other potential occurrences of this type of coupling during SH winter months. SAO is known to occur regularly, but with appreciable inter-annual variability (Smith et al., 2020). The QBO on the other hand has an average

period of 28 months.

#### 3.3.1 SAO-QBO interaction in 1988 and 2017

In our analysis of individual years we found wave-mean flow interactions similar to the cases of 2019 and 2002 taking place on two additional austral winters: 1988 and 2017.

The winter of 1988 has a close resemblance to 2019, but the SAO-QBO interaction occurs about a week later, with similar,

sectoral wind reversal patterns to 2019 delayed to late September in 1988. Figure 8 is as Figure 3, but for the austral winter of 2017. Figure 8a shows the easterly SAO in the mesosphere and the QBO in the stratosphere. Poleward of the SAO in the mesosphere, EP flux convergence similar to 2002 and 2019 is taking place. The SAO and QBO features intersect earlier, between June 29 - July 5, creating an extended zero wind line at 30°S from 40 hPa to 0.3 hPa, Figure 8b. As before, the zero wind line subsides in August, Figure 8c, extending now from 40 hPa to 3 hPa. Meanwhile, easterly momentum continues to be

deposited in the mesosphere throughout these winter months, but this is does not extend as far into the stratosphere as in 2002 and 2019. This likely leads to delays in the wind reversal, which now takes place during October 23-29, Figure 8d.

#### 3.3.2 Remaining eQBO years

Here, we present similar analysis to Figures 3 and 6, however, instead of individual years, the averaging is now based on the MERRA-2 streams (Bosilovich et al., 2016) described in section 2.1. Note that all years were initially analysed individually –

the stream averages were found to be representative of the individual years, and no SSW like behaviour was observed for the individual years.

Figure 9 shows the zonal wind, EP flux and EP flux convergence averaged over June 15-21, July 6-12, August 10-16 and August 31 - September 6, averaged for the years 1980, 1983, 1985 and 1990. In Figures 9a and b, the SAO wind pattern is noticeable, but not as distinct as before, above 1 hPa. Figures 9c and d, show how the easterly QBO evolves, but no noticeable

easterly momentum is deposited. No SAO-QBO interaction is taking place during June-July. For 1980 and 1983 we find the



SAO-QBO interaction taking place, but much later in winter, late July/early August, compared to 2002 and 2019, and the zero wind line in both cases does not extend into the mesosphere, leaving the wave propagation pathway to the equator accessible.

Figure 10 is the same as Figure 9 but now averaged for the years 1993, 1995, 1997 and 1999. The results are very similar to Figure 9; Figures 10a and b show the SAO in the mesosphere in June and July. Figures 10c and d show the enlargement of the eQBO wind pattern and the return of westerlies to the Northern Hemisphere. None of the individual years displayed a SAO-QBO interaction similar (with regards to timing and zero wind line extent) to 2002 and 2019 at any point during winter. Figure 11 is the same analysis but now averaged for the years 2004, 2006 and 2008. Figure 11a now shows the SAO in the mesosphere in June, similar to 2002 and 2019. The SAO and QBO show signs of early interaction, but, this is not sustained (Figure 11b) in late July. This results in a zero wind line between roughly 10 and 1 hPa, smaller vertical extent than the two in 2002 and 2019. Figures 11c and d show eQBO enhancement and the return of westerlies to the NH as austral winter fades into spring.

The average for the remaining eQBO years, 2011 and 2014, is shown in Figure 12. In early July (much later than the other years analysed) we find a weak SAO signal at 1 hPa (Figure 12a), and it does not extend further into the SH as happened in 2019 and 2002. Thus no early winter momentum deposit takes place in the winter mesosphere. The SAO and QBO intersect (Figure 12b) in late July, triggering easterly momentum deposit in the polar mesosphere. However, much like in Figure 11, this SAO-QBO pattern is not pushed beyond 30°S, unlike during the SSW years. The QBO stays enhanced into August (Figure 12c). The QBO eventually subsides into the lower stratosphere in September, Figure 12d.

In general, the QBO and SAO both appear in the years analysed here, however, their vertical and poleward extents vary. This seems to influence the timing and extent on the SAO-QBO interaction. The two SSW years, 2002 and 2019, have a large (vertical and poleward extent) eQBO at 10hPa, and a large (vertical and poleward extent) SAO in the mesosphere before their intersection in July, which produces a zero wind line stretching from the stratosphere into the mesosphere. The SSW-like years of 1988 and 2017 have similar early winter behaviour to 2002 and 2019, however, mesospheric wind reversal takes place later. The 1980s and 1990s were characterised by having both a smaller QBO and SAO which only occasionally interacted in late winter. Whilst the 2000s had a smaller QBO and larger SAO, which interacted but did not produce a zero wind line similar to the SSW years. The 2010s had a large eQBO, with a smaller SAO which did interact in July, however, no SSW was produced in September. This suggests that not only may the particular phases be important for preconditioning the area for a SSW, but there vertical and poleward extents (and thus any mechanisms influencing these) seem to also be a factor.

## 4 Discussion

The sudden stratospheric warmings over Antarctica in 2002 and 2019 both have an early winter equatorial SAO-QBO interaction and coinciding easterly momentum deposits in the polar mesosphere. For both years the SAO is a clear feature of easterly winds extending into the SH before it reaches 1 hPa. This shift from (polar) westerly to (low latitude) easterly winds changes the waveguide in the mesosphere, which results in easterly momentum being deposited in the mesosphere from early winter. When the QBO and SAO easterly wind features merge, they generate a zero wind line that stretches from the lower stratosphere



into the mesosphere near 30°S latitude, now modulating the wave guide across the whole vertical range. This QBO feature

then continues extending towards the pole into August, with continued easterly momentum deposit in the polar mesosphere, decelerating the mesospheric westerlies. By September, the extended momentum deposit results in reversal of mesospheric winds. The year 2002 saw the zonal mean zonal winds decelerate down to below 10 hPa. However, in 2019, the mesospheric zonal mean zonal wind reversal links across latitudes from the pole to the equatorial QBO. In our zonal mean analysis, this suggests that vertical wave propagation from the source regions at high and mid-latitudes is significantly affected, with the

movement of the zero wind line creating a barrier for upwards propagation.

The SSW-like years of 1988 and 2017 show similar SAO-QBO interaction in July. Whilst the winds did reverse in the mesosphere in late September-October (later than we found for 2019 and 2002), there does not appear to be a rapid warming similar to 2019 and 2002. Although, in 2017 the changes in dynamics were enough to stifle the growth of the ozone hole (Klekociuk et al., 2020). Causes of these differences should be investigated further in a detailed study of these SSW-like years.

Our analysis of all other years with similar background QBO conditions did not reveal similar behaviour with early winter momentum deposit and similar SAO-QBO interaction. Hence we proceeded to further analyse the other eQBO years by decades of matching MERRA-2 streams. The general behaviour of the polar atmosphere during eQBO in the 1980s and 1990s was similar. In general, both decades show a weak signature of the SAO in the mesosphere. However, the SAO-QBO interactions were either: later in the season (August) or did not result in a clear poleward shift of the zero wind line. We did not find evidence

of easterly momentum being deposited throughout the winter as we did for 2002 or 2019. The later decades, the 2000s and 2010s, did in general have a SAO present in the mesosphere in early winter and the SAO and QBO did interact in July, similar to 2019. However, this did not result in continued deposition of easterly momentum through the winter, or mesospheric wind reversals in September.

Recently, Gray et al. (2020) reported that in order to accurately simulate NH SSWs in an atmospheric model, not only was

is necessary to constrain the model's global tropospheric winds and temperatures, but further constraining of the zonal wind in the equatorial atmosphere above 5 hPa to reanalysis fields was also needed. These model results further emphasize those of Gray (2003) who showed similar results for a middle atmosphere only model: that the high altitude equatorial atmosphere plays an important role in NH extreme events. Although our analysis focuses on the SH SSWs in 2002 and 2019, our results present a possible physical mechanism for this connection. The early winter SAO-QBO interaction and subsequent modulation

of the wave guide reflects mid-latitude waves up and pole-ward, resulting in easterly momentum deposit in the mesosphere. The SAO-QBO interaction is not unique to 2019 and 2002 and was found to happen during other easterly QBO years. However, apart from the two SSW-like years of 1988 and 2017, the timing and extent of the zero wind line was not found to occur in these other years. We suggest that this may be a reflection of variations not only in QBO but also in the amplitude and descent pattern of the SAO, the latter of which, to our knowledge, are not well understood (see e.g. Moss et al., 2016; Kawatani et al.,

315    2020).

We propose that this early winter behaviour may be a key physical process in decelerating the mesospheric winds, which may precondition the atmosphere for a SSW. It may also help explain why SSWs are less common in the Southern Hemisphere: the early and large SAO-QBO interaction is dependent on both the QBO being in the correct phase, and the SAO appears to need





to have a large amplitude and descend down into the stratosphere during the early-mid winter. As we find that the early key
patterns start occurring 2-3 months before the SSW event, the behaviour of the equatorial middle atmosphere along with the
polar response at this stage may heed an imminent SSW event, potentially providing predictability beyond the current 20-30 day
window in SSW prediction on both hemispheres (Lawrence and Manney, 2020; Rao et al., 2020). This in turn could potentially
aid subseasonal to seasonal (S2S) prediction (Domeisen et al., 2019a). However, this would need to be tested in detail for the
NH atmosphere. We note that the results presented by Gray et al. (2020) (their Figures 3-4) suggest that coupling of the SAO
and QBO zonal wind patterns, similar to our SH cases, took place in their simulation approximately 2 months before the onset
of the NH January 2009 SSW. As noted by Gray et al. (2020), the atmospheric region where the SAO originates (mesosphere),
tends to be neglected in model development. Our results provide further evidence that these mesospheric altitudes are not only
important for understanding the NH but also the SH extreme dynamical events.

As mentioned earlier, much work has been done in understanding both causes and implications of SSWs, particularly in the
NH. Baldwin et al. (2021) provide a recent review of the current understanding, including the many interactions with large
scale atmospheric modes that have been found to influence NH SSW occurrence, including the QBO, the ENSO, and the MJO.
Due to the scarcity of SH SSW events we were unable to investigate the potential individual influences of these. However,
we note that ENSO conditions, based on the Multivariate ENSO Index (MEI.v2, Zhang et al. (2019) available at https://psl.
noaa.gov/enso/mei/, last accessed 3 Dec, 2020), were neutral, while the MJO index (Wheeler and Hendon (2004), available at
https://www.cpc.ncep.noaa.gov/products/precip/CWlink/MJO/whindex.shtml, last accessed 3 Dec, 2020) was positive, during
both 2002 and 2019. For the SSW-like years of 1988 and 2017 the MJO was generally variable during the austral winter, while
the ENSO index was negative, and thus opposite to the two generally recognised SSW years. These difference could signal the
importance of teleconnections in the SH polar responses.

## 5 Conclusions

Sudden stratospheric warmings are disruptions to the seasonal cycle of the polar winds. Only two have occurred over Antarc-
tica: one in 2002 and another in 2019. Here we present results based on the MERRA-2 reanalysis, showing that during both
years, 2002 and 2019:

1. From early winter, waves are depositing momentum in the polar mesosphere in a manner consistent with the equatorial
   SAO pushing the wave guide boundary into the SH.

2. An early winter SAO and QBO interact in the equatorial atmosphere, driving further momentum deposit, and thus zonal
   wind deceleration, in the polar mesosphere.

3. Changing zonal winds further influencing wave propagation conditions, ultimately transferring the signal to the strato-
   sphere and triggering SSW conditions.


For both years the SAO is pronounced before it reaches 1 hPa, and easterly momentum is deposited in the mesosphere before
the intersection, and is sustained throughout the season. When the equatorial eQBO and SAO wind patterns interact they result
in a zero wind line that stretches from the lower stratosphere into the mesosphere near 30°S, modulating the wave guide.

Previous work focused on the more frequent NH SSWs has pointed to the role of the equatorial upper stratosphere and
mesosphere particularly when simulating the timings of NH SSW (Gray et al., 2020). Our analysis of the two SH SSWs
suggest that, at least for the Southern Hemisphere, the interaction of the QBO and the SAO in the equatorial upper stratosphere-
mesosphere seems critical in triggering the polar disturbances. The SAO-QBO interactions in 2002 and 2019 are not unique,
however, the coupling and extent of the equatorial wind patterns occur much earlier in the season and result in a longer
zero wind line when compared to most other easterly QBO years. We propose that this early winter behaviour may be a
key physical process in decelerating the mesospheric winds, leading to preconditioning of the polar atmosphere for a SSW.
The occurrence of these patterns in the equatorial atmosphere and in the polar mesosphere during early winter could provide
extended predictability of SSWs from the current 20-30 day window (Lawrence and Manney, 2020; Rao et al., 2020) to 2-3
months. As our present analysis is based on SSWs that took place in the SH, further work would be needed to test to what
extent SAO-QBO interactions might play a role in NH SSWs. This future work may help shed light on the different roles
background flow and wave enhancement have on triggering the SH and NH SSWs, which may help explain why SH SSWs
occur less frequently than NH SSWs.

*Data availability.* Global Modeling and Assimilation Office (GMAO) (2015), MERRA-2 inst6_3d_ana_Np: 3d,6-Hourly,Instantaneous,Pressure-
Level,Analysis,Analyzed Meteorological Fields V5.12.4, Greenbelt, MD, USA, Goddard Earth Sciences Data and Information Services
Center (GES DISC), Accessed: [21 October 2020], 10.5067/A7S6XP56VZWS.

The QBO phase information from MERRA-2 is available from https://acd-ext.gsfc.nasa.gov/Data_services/met/qbo/ [last accessed 24
November 2020]. The QBO phase information from daily sonde measurements is available at https://acd-ext.gsfc.nasa.gov/Data_services/
met/qbo/QBO_Singapore_Uvals_GSFC.txt [last accessed 19 November 2020].

The ENSO Index, MEI.v2 (Zhang et al., 2019) is freely available at https://psl.noaa.gov/enso/mei/ [last accessed 3 Dec, 2020].

The MJO index (Wheeler and Hendon, 2004) is freely available at https://www.cpc.ncep.noaa.gov/products/precip/CWlink/MJO/whindex.
shtml [last accessed 3 Dec, 2020].

*Author contributions.* AS and VJN planned the study. VJN analysed the MERRA-2 data with comments from AS. Both authors contributed
to the writing of the article.

*Competing interests.* The authors declare no competing interests.



*Acknowledgements.* We are grateful for the open access to the MERRA-2 reanalysis data products provided by GES DISC.

The work of VJN was supported by a postgraduate scholarship provided by the University of Otago Physics Department.



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





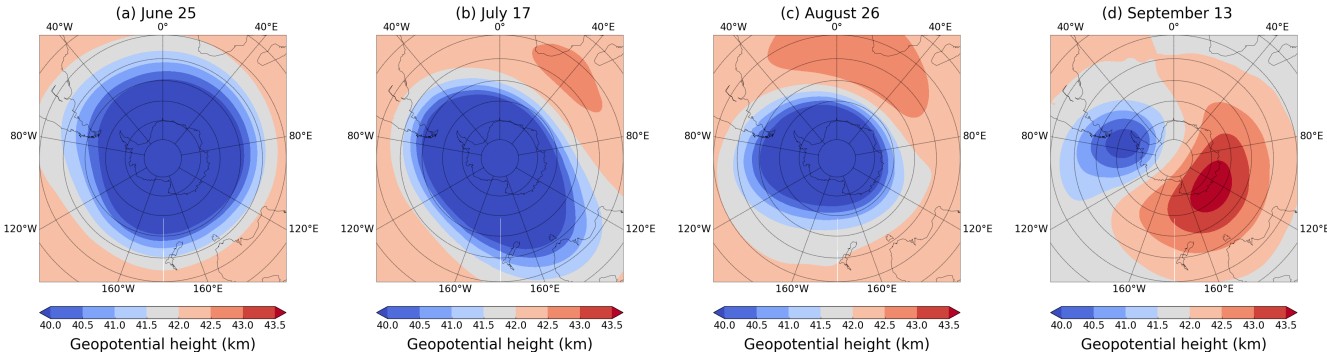

**Figure 1.** MERRA-2 geopotential height maps for 2 hPa pressure level over Antarctica for selected dates during winter 2019. Low geopotential heights indicate the position of the polar vortex. Latitude circles are shown at 10° intervals and longitudinal sectors are shown at 40° intervals. Dates shown in individual panels are (a) June 25, 2019, (b) July 17, (c) August 26, (d) September 13.

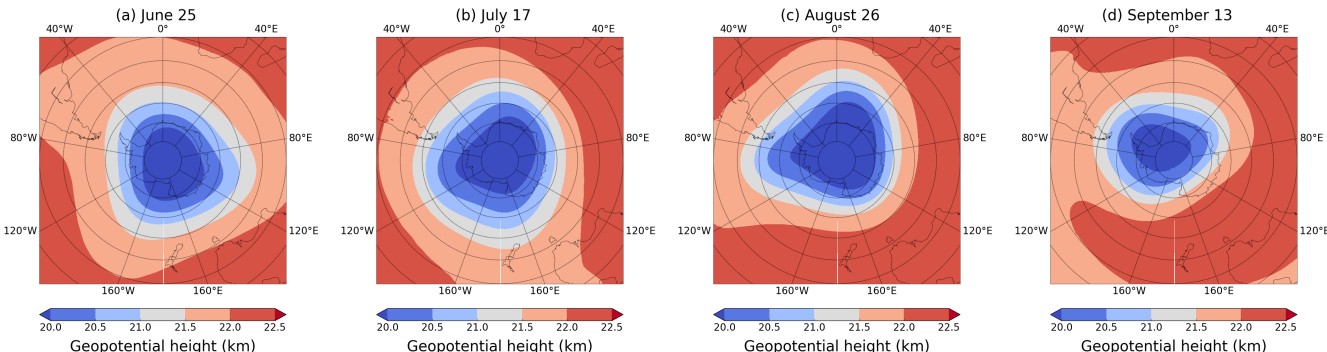

**Figure 2.** As Figure 1 but for the 40 hPa pressure level.



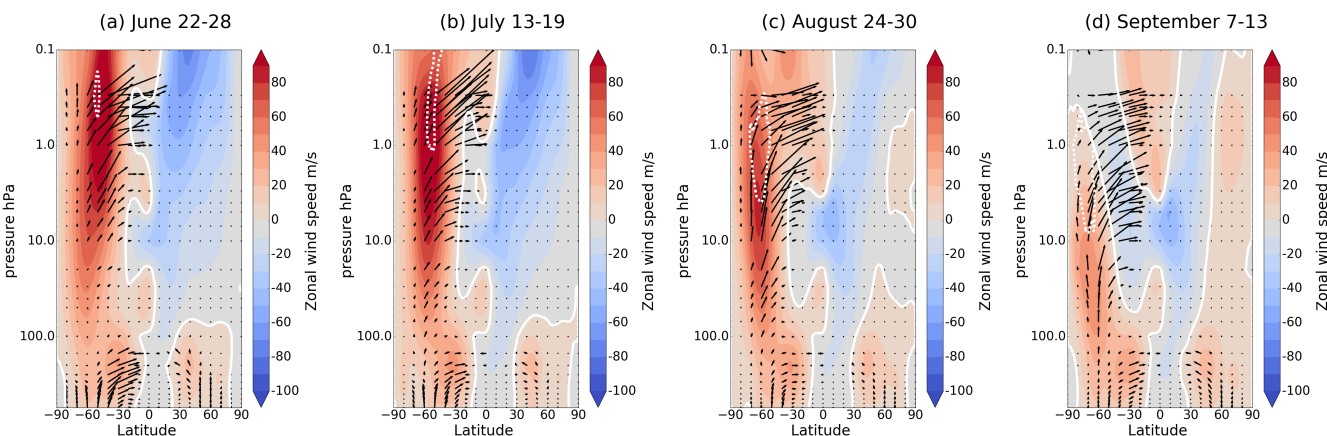

**Figure 3.** 7 day average zonal mean zonal wind (filled contours, ms$^{-1}$) and Eliassen-Palm (EP) flux (arrows, m$^2$s$^{-2}$) for the austral winter 2019 from MERRA-2. Each figure covers the latitudinal range of 90°S to 90°N and the vertical range of 550 hPa to 0.1 hPa. The time periods (exact dates as given in figure titles) shown have been selected to depict the evolution of the events. Note that the time periods encompass those shown in Figure 1. EP flux convergence is indicated with dashed line. The solid white line shows the location of the zero wind line.

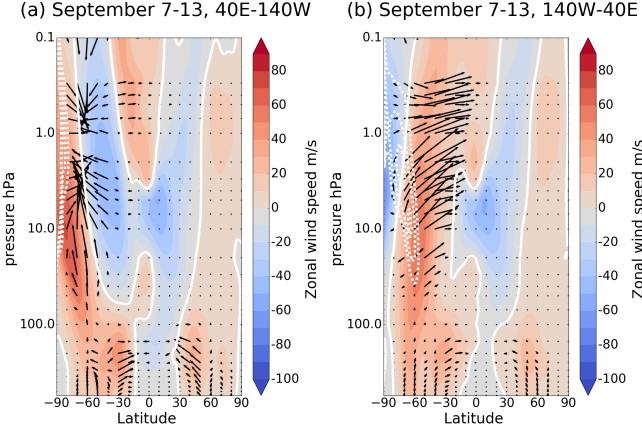

**Figure 4.** The same as Figure 3 (d), but the zonal averaging in EP flux calculation according to equations (1) and (2) has now been made over the longitudinal sectors (clockwise) (a) 40°E-140°W and (b) 140°W-40°E .

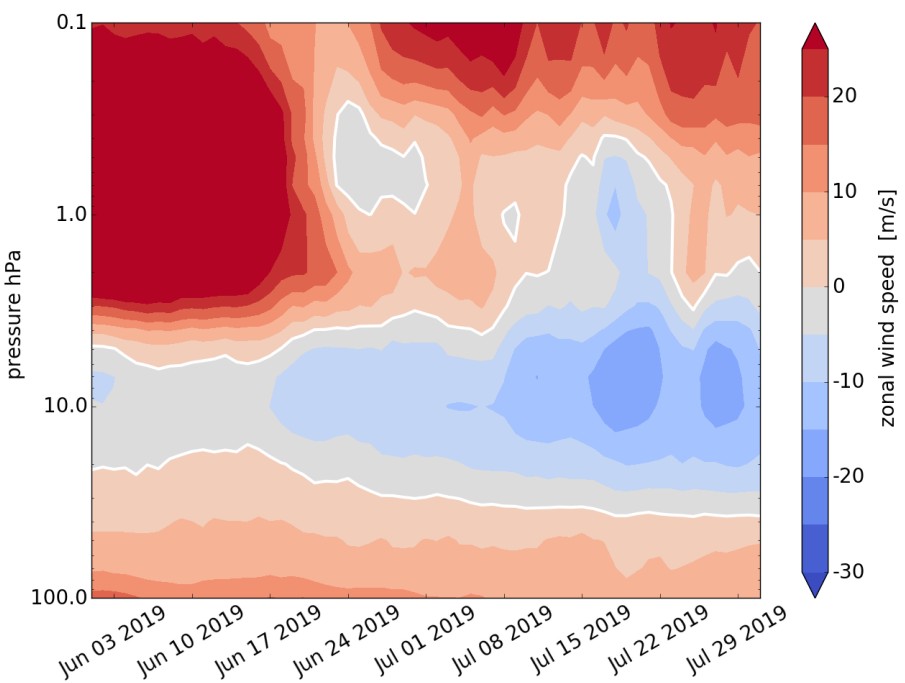

**Figure 5.** Temporal evolution (June 1 to July 31, 2019) of the zonal mean zonal wind (ms$^{-1}$) in the vertical range of 100 hPa to 0.1 hPa, averaged between 15-20°S. The white line signifies the zero wind line.

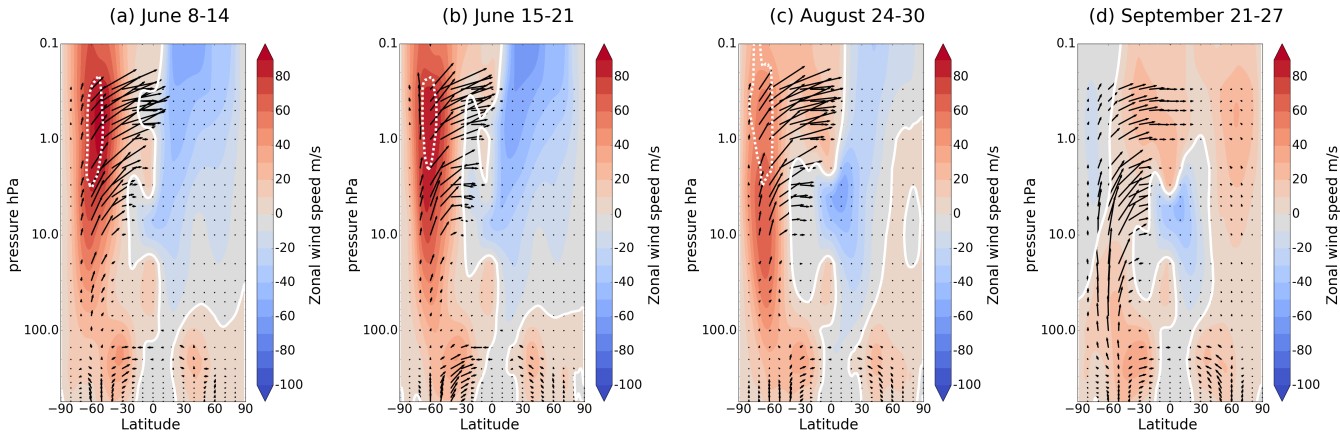

**Figure 6.** As Figure 3 but for the year 2002. The time periods (exact dates as given in figure titles) shown have been selected to depict the evolution of the events. Panel (d) corresponds to the SSW and vortex split event.



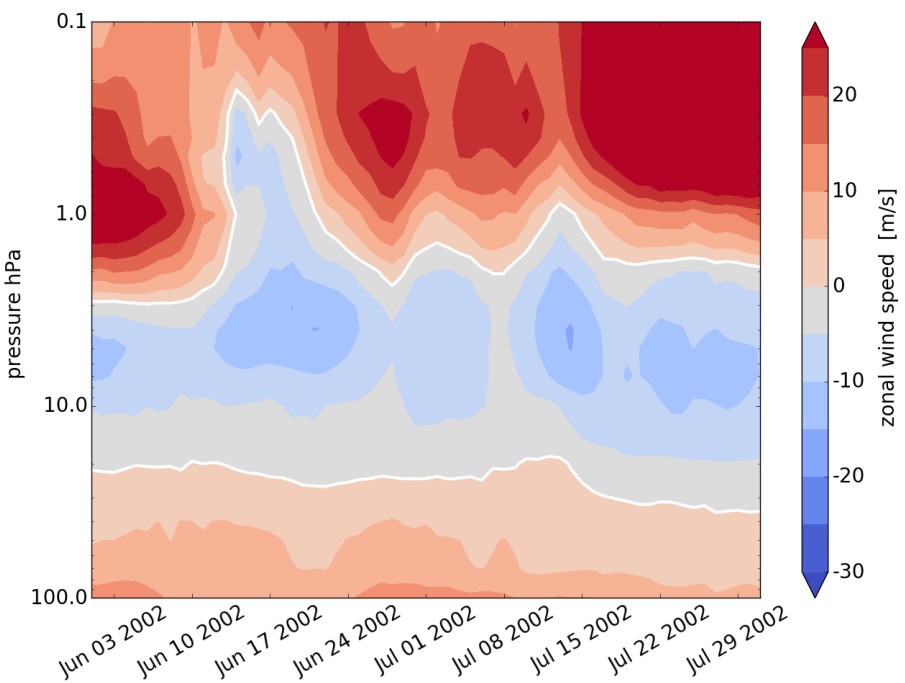

**Figure 7.** As Figure 5, but for the year 2002 .

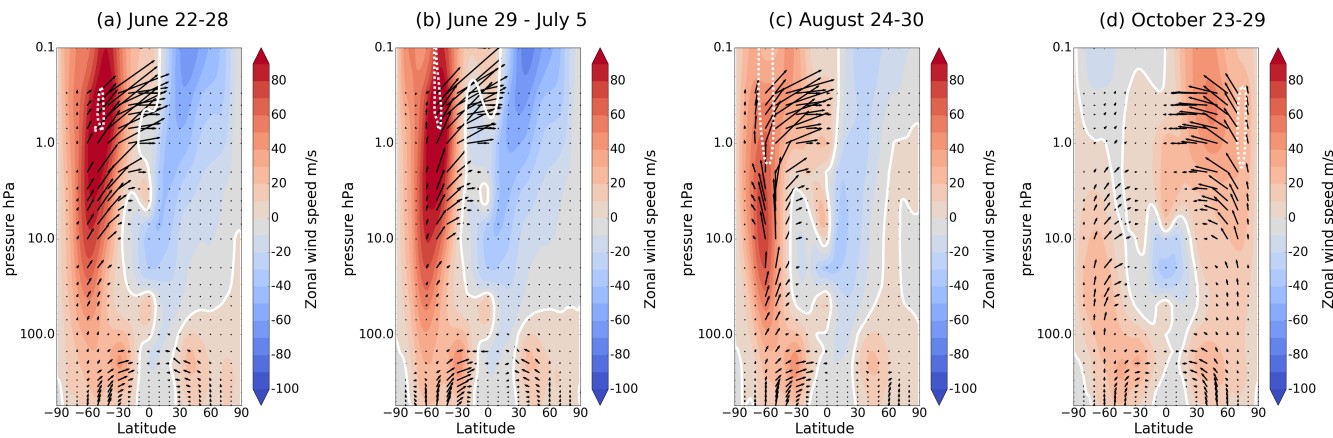

**Figure 8.** As Figure 3 but for the year 2017. The time periods (exact dates as given in figure titles) shown have been selected to depict the evolution of the events.





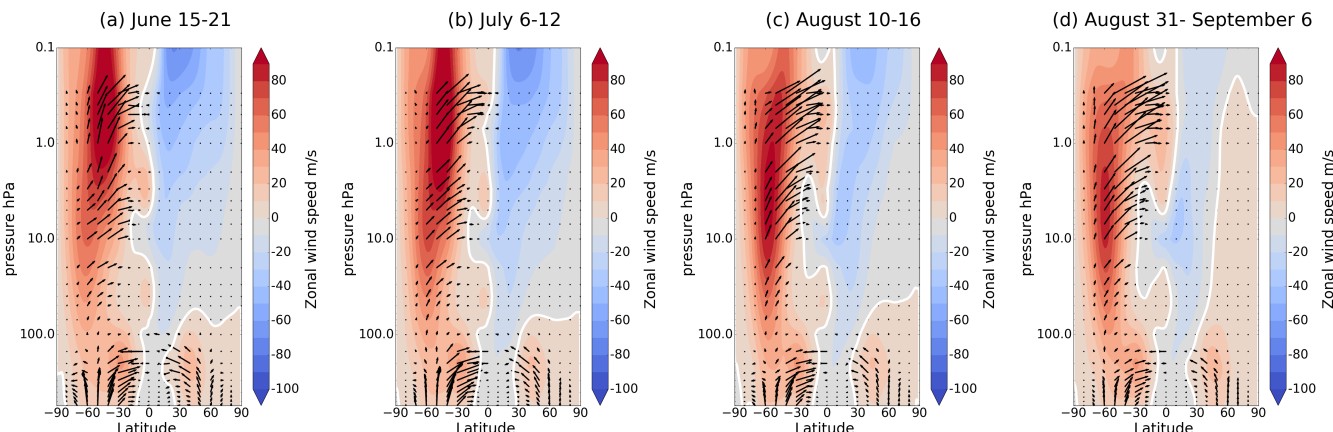

**Figure 9.** As Figure 3 but averaged over 1980, 1983, 1985 and 1990. Time periods for the 7 day averages as given in the figure titles.

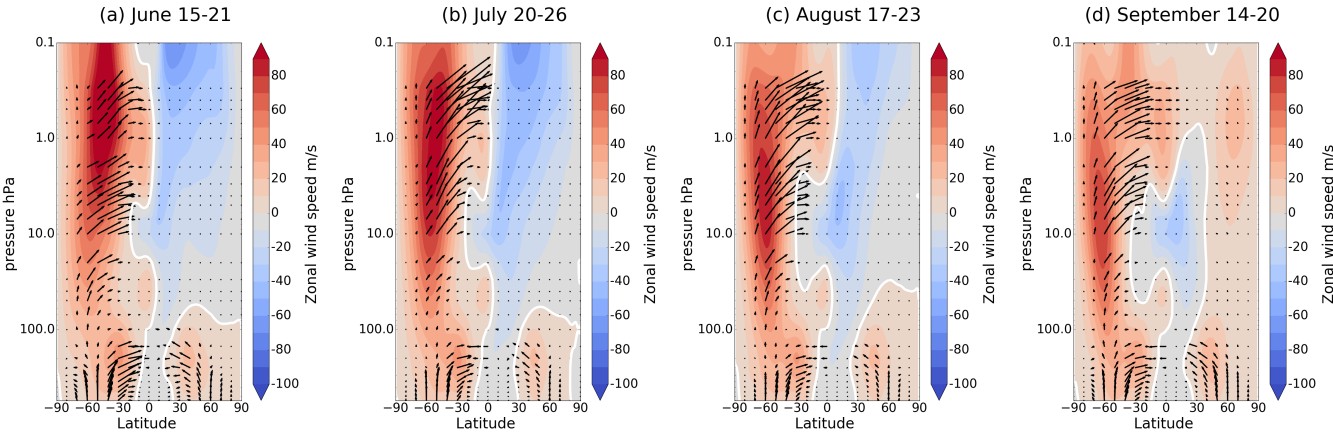

**Figure 10.** As Figure 3 but averaged over 1993, 1995, 1997 and 1999. Time periods for the 7 day averages as given in the figure titles.





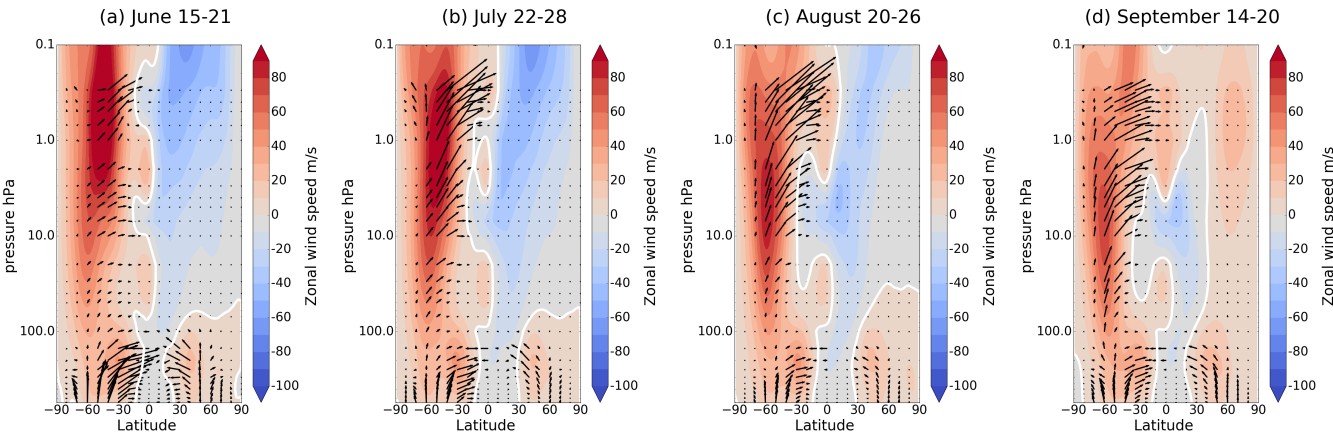

**Figure 11.** As Figure 3 but averaged over 2004, 2006 and 2008. Time periods for the 7 day averages as given in the figure titles.

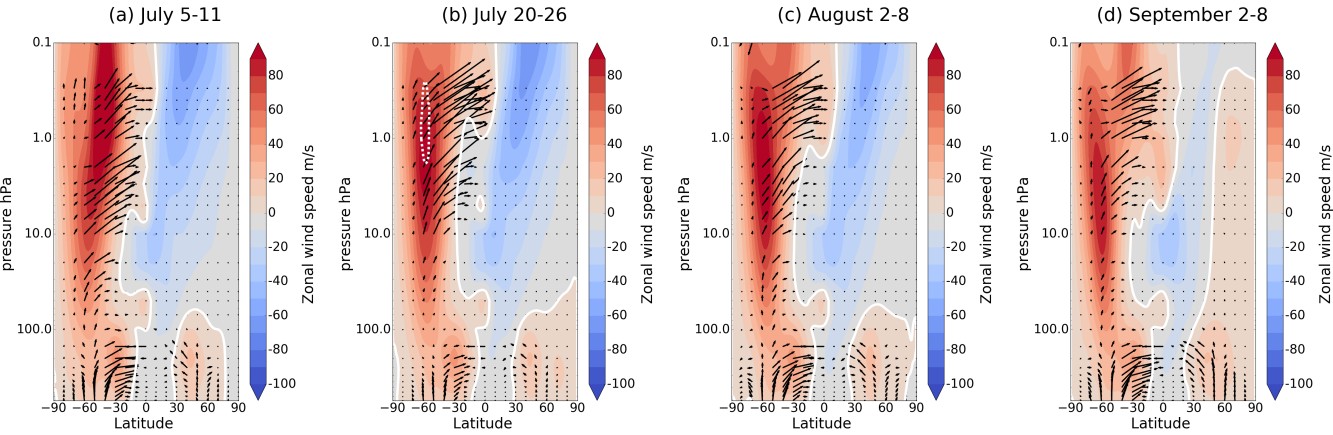

**Figure 12.** As Figure 3 but averaged over 2011 and 2014. Time periods for the 7 day averages as given in the figure titles.