# Peer review of "Does the coupling of the mesospheric semiannual oscillation with the quasi-biennial oscillation provide predictability of Antarctic sudden stratospheric warmings?"

_Atmospheric Chemistry and Physics, 2020_

## Author Comment (AC1)

We would like to thank all reviewers for their comprehensive comments on our manuscript. We performed additional analysis as suggested by the reviewers. The new results have made the results clearer and aided in providing more quantitative results through the manuscript.

We note that following the feedback we revised all figures for clarity and removed several following suggestions as these were not necessary for the paper.

In addition, the stream based analysis has been revised to include all reference years together, which has clarified the paper and allowed us to add two new figures presenting the 2002 and 2019 anomalies from this reference.

**Responses to Reviewer #1**

In their paper, Nordström and Seppälä present a coupling of SAO and QBO phases that has occured both during the 2002 and during the 2019 Antarctic SSW events. The authors describe some characteristics of the two wind phenomena that, in combination, were special in these two years and hence may be crucial for the generation of the SSW events. In other years of similar QBO and SAO phase combinations, some differences can always be pointed out, but the method of doing so is not convincing. The conclusions of the paper are all in all pretty vague, and I think they could, with little effort, be concretised such that a clear statement stands at the end of the paper. I think the study presents an interesting mechanism that has the potential to add a fair bit of understanding to the SSW phenomenon and I recommend publication in ACP. However, there are a couple of major and numerous minor points that I would like to see addressed beforehand and also the presentation quality requires some improvement, especially because the language is often too unprecise. Please consider the list of major, minor and technical issues below for that.

**Major issues**
- I do not understand how it can be fair in this study to compare in Figs. 9-12 an average over several years with particular years. This way you can average out particular features of certain years or even create artefacts in the averages. The dates that are chosen appear rather arbitrary and not well defined. If they are specific for each year (which seems to be the case for the two SSW years), how can they fit to the averages over several years? I also did not really undestand how it makes sense to put these particular years into one pot (the use of the 'streams'). I guess this requires some good explaining or serious revision of the method.

   **Response**: We have revised all these types of figures and now show the full temporal development during the winter by including all time periods as separate panels, rather than only showing selected time periods. We also revised the analysis that was split into streams and provide a reference of the eQBO years that is presented in a similar way, showing the temporal evolution over the winter months.

   We agree that averaging over several years could lead to averaging out particular features, which is why we checked all the years individually. We did not find evidence for similar behaviour, as was mentioned originally, which brings some reassurance for the use of a reference average over the other eQBO years. While we don't think it would be feasible to include the figures for all the individual years in the paper, these could be added in as a supplement.

- In some of the additional years, the SAO-QBO interaction takes place, but then the signal is not moving down all the way. Too often in the paper, it is stated that eQBO and eSAO come together, but then this does not result in change in deposition of wave forcing/wind reversal/... (see 301-302 as example) but then, the reason for that is not discussed. This happens throughout the paper, it always leaves the reader behind puzzled with nothing in his hands. Some more analysis or discussion to at least point out the one or the other mechanism that could be responsible for this would be very helpful.

**Response**: Following the reviewer comments from all reviewers we performed more analysis on the behaviour of equatorial winds as well the location of the zero wind line. We now show some more compelling evidence that early winter information on the location of the 1 hPa zero wind line together with EP flux convergence in the polar vortex edge region in the upper stratosphere is able to identify the SSW/early onset weak vortex events during easterly 10 hPa QBO winters.

- In the analysis, gravity wave (GW) drag is completely missing. GWs can also largely contribute to the overall wind forcing in these areas. These data should be available from MERRA-2 as well. That should be added to the resolved wave drag and the contours should be plotted more nuanced. As is, the caption does not even provide information on the values of the drag and there is only one (or no) contour line. Possibly, clearer conclusions can be drwan on the wave activity statements with this done.

  **Response**: We have revised all the figures and now include $\pm 2$ m/s/day contour lines in all figures with EP flux divergence.

  We have not added a detailed analysis on gravity wave drag as this would have added much more material to the manuscript, but this should be considered in any following work.

- The paper fails to make any quantitative statements about the events. In consideration of the few examples, it is clear that this cannot be made in detail. However, some more effort could be made to achieve something in that direction. One could maybe make statements about when it is too late in winter for the two signals to fall together for generating an SSW as seen from example years where this happened. Or the magnitude that is reached could be described better. For example, in L269-270 a 'large' eQBO and SAO is mentioned but not even stating what 'large' means. Is it the strength or the expansion of the easterly winds that matters? With statements that are a bit clearer, the conclusions of the paper could step up a little from what is now stamped by a conditional that reflects in the word "may" too often (see e.g. L316-). The mechanism that is described in the paper is quite clear and interesting, so it is a little disappointing that at the end the conclusions are so vague. I think this could be improved fairly easily for example by some statistical analyses.

  **Response**: We have added new analysis based on reviewer suggestions (please see figures included later in the responses) and now provide much more quantitative information throughout. We added analysis including looking at the timing of the wind pattern development together with EP flux divergence response, based on which we suggest that the early winter period from mid-June is critical for the SSW events.

**Minor issues**

- L10: "these features". Please name them.

  **Response**: Revised to "As the winter progresses, the momentum deposition and wind anomalies descend into the stratosphere,…"

- L28: When did record keeping begin?

  **Response**: To our understanding there is scarce information before the 1980s. We have removed this sentence.

- Revised to "The 2019 SSW is one of two that have been recorded to have taken place in the SH…"

**Response**: This now reads:

*"The 2019 SSW is one of two that have been recorded to have taken place in the SH (Rao et al., 2020c), and provides a unique opportunity to investigate the atmospheric conditions leading up to SH SSWs."*

- L31-32: This is a bit oversimplified for my taste, please don't forget about PSC processes here. (below you mention it then) Moreover, use the common wording 'ozone depleting substances'. These substances are there, no matter if "trapped" or not. Please remove the word "trapped".

  **Response**: This was meant to only signify why some of the influences of SSWs are important. We have revised this to:

  *"For example, SSWs contribute to the size of the ozone hole via two different mechanisms. First, the warming of the stratosphere suppresses the formation of polar stratospheric clouds (Shen et al. 2020), which play a critical part in stratospheric ozone depletion (Solomon, 1999). Furthermore, the weakening of the vortex allows the mixing of ozone rich mid-latitude air into the pole. Both these effects in combination lead to a smaller ozone hole (Solomon et al., 1986)."*

- L55: 'at higher altitudes' and 'at lower altitudes' in one sentence confuses me. Please rephrase.

  **Response**: As the figures were removed following reviewer suggestion, we also removed this text.

- L61: Rephrase to: Hence, the dynamic situation in 2010 was unlike the situations in 2002 and 2019, because in the latter two years, rapid warmings and wind reversals occured.

  **Response**: Revised as suggested.

- L96: What does 'well' mean here? Realistically or at all or …?

  **Response**: realistically. We have revised this text to state this.

- L104-109: I do not think this kind of summary belongs here, rather state how the paper is structured at this point.

  **Response**: We removed this as suggested.

- For my taste, the introduction (and also the methods section) is too long. I do appreciate the detailed information, but I think the authors could try to somewhat compress it. Sometimes it is a bit of a back and forth and moreover, I am not sure if all the information is really needed for the paper.

  **Response**: Considering this comment we reduced the text in both sections so they are now shorter. We hope that this helps clarify the manuscript. However, we note that another reviewer requested adding significantly more material to the introduction.

- L136: Remove: 'in zonal winds'

  **Response**: Revised as suggested.

- Fig. 3 caption: Denote the magnitude of the EPfc line!

**Response**: We have revised the figures that depicted the EP flux and EP flux divergence and now include the magnitude.

- L160: cite here: Middle atmosphere dynamics, Andrews, David G and Holton, James R and Leovy, Conway B, 40, doi:10.1002/qj.49711548612, 1987, Academic press

  **Response**: We removed the equations following a request by a reviewer.

- L195-196: "reverse further down than the mesosphere" I do not understand what you want to say here. Can you rephrase that please.

  **Response**: We have revised this section to include more quantitative description.

- L200-2005: Rephrase to: In Fig. 4, we average Fig. 3d according to the two geopotential anomaly patterns from Fig. 1, namely clockwise from 40..... This way, we find …. But actually, do you conclude anything from these figures? Or will you refer to this lateron? If so, state it. If not, you can remove this bit, because it shows simply what you would expect.

  **Response**: We removed this as suggested.

- L206-209 and associated figure: That is a nice depiction too, but not needed for understanding. If no conclusions are drawn from this here, move it to where it is needed, or remove it. (but see below)

  **Response**: We took the advise and removed this and the other similar type figure, as they were not necessary.

- L218-219: rephrase 'comes to lay' and please the entire sentence, it is very confusing

  **Response**: This text was removed.

- L221: Rephrase to: Fig. 6d shows that the polar winds revers between …

  **Response**: We have revised all text in this section.

- L223-225 If the purpose is to contrast Fig. 5 and Fig. 7, show the two figures together here and describe differences. Wouldn't it be interesting to extend these time series to when the SSWs happen to see the entire evolution? Or is not so much happening in these latitudes then?

  **Response**: With the addition of new analysis requested by reviewers, and suggestion that these figures were not adding much, we ended up removing both figures.

- Remove subsection headers 3.3.1 and 3.3.2

  **Response**: Removed.

- L235: Fig 8 is as.... remove the sentence

  **Response**: Figure 8 was removed.

- Fig. 9-12: I cannot see any EP flux convergence contours in these plots (exception is 12b). I can hardly imagine that there is none anywhere there. So how come?

  **Response**: We have made these contours clear in the new revised figures.

- L249: 'as before' What are you referring to exactly? Be precise.

**Response**: This section was removed in the revision process following reviewers suggestions.

- L256: How can you expect the timing to be so similar to 2002 or 2019, when even the timings of 2002 and 2019 are not really similar. How similar does it have to be?

  **Response**: This section was removed, but overall, we now explore the timing more by detailed investigation of the development of the equatorial wind patterns and how the location of the zero wind line changes.

- L273-275: 'smaller' and 'larger'. In what sense? Expansion, or strength, both? Be specific.

  **Response**: This section was removed in the revision process following reviewers suggestions.

- L273: Rephrase to: ...QBO and SAO and the two patterns only occasionally interacted…. But actually, I do not like this statement at all. How can you summarise over whole decades, when each year is so specific and has its very own dynamics and each feature can be of importance. This refers to my main point, why I think the averaging over several years is not feasible in this study.

  **Response**: As pointed out by all reviewers this averaging method was not suitable and we have revised the results and use a reference mean of all eQBO years without early onset vortex weakening events. All text has been revised accordingly.

- L280: Do you mean: In both years, the SAO is in its easterly phase with winds extending into the SH above 1 hPa... ? (also L349)

  **Response**: We have revised this to:

  *"For both years the SAO is presents as a feature of easterly winds (of over 10 ms⁻¹) extending into the SH from early winter."*

  We also now show and discuss the SAO winds in context of their magnitude in other years.

- L287: The 'However' here confuses me. Do you want to express 'in contrast'? But I don't see how these two situations stand in contrast. Can you resolve this?

  **Response**: We have clarified this contrasting, along with the revised figures, and now write

  *"In 2002, the zonal mean zonal winds between about 40S-60S decelerate down to below 10 hPa, eventually triggering major SSW conditions. In contrast, in 2019, the zonal mean zonal wind reversal is less focused, taking place across a wider range of latitudes, and major SSW conditions are not fulfilled."*

- L288: What does 'links across latitudes' mean?

  **Response**: We have clarified this as shown above.

- L308: I guess you mean stratospheric extreme events? As is it appears like referring to troposphere. Better simply write SSWs.

  **Response**: This was revised to say SSWs as suggested.

- L317-320: The reason that SSWs are less common in the SH is not explained here! I reckon the eQBO and eSAO co-occur about as often in both hemispheres, and then the wave guide is changed similarly in both hemispheres. But what is different is the amount waves that then uses

that waveguide and makes its way up to the mesosphere. In the SH there are generally much less/weaker waves and hence the perturbation in the upper atmosphere is weaker.

**Response**: We have clarified the text here to:

*"We propose that this early winter behaviour may aid in identifying conditions that lead to deceleration of the polar winds, which could then precondition the atmosphere for a SSW. For example in 2019, when there was enhanced upwards wave flux in August (Shen et al., 2020), the modulated waveguide in the stratosphere and above may have provided further optimal conditions for large disturbance to take place. It may also partially explain why SSWs are less common in the Southern Hemisphere: if the early and large SAO-QBO-like merging contributes to optimal conditions for SSW, not only is this dependent the QBO being in the correct phase, but also the SAO having a large amplitude during the early-mid winter. The SH winter typically experiences smaller amplitude SAO easterlies, while the NH winter experiences much larger easterly winds (Smith et al., 2017). However, this hypothesis would need to be tested separately for the NH. It is also important to note again that the NH has higher winter planetary wave activity and variability than the SH."*

**Technical issues**

- L30: What do you mean by 'descend' here? I guess a different word could be more suitable.

  **Response**: This has been revised to "During the polar winter,…"

- L21: Change 'aftermath' to 'influence'

  **Response**: Revised as suggested.

- L59: Change 'had' to 'included'

  **Response**: Revised as suggested.

- L66: Change 'However' to 'In contrast'

  **Response**: Revised as suggested.

- L67: remove 'flat' and change 'don't' to 'do not'

  **Response**: Revised as suggested.

- L83: change 'a switching' to ' and 'alternation'

  **Response**: Revised as suggested.

- L96: change 'found' to 'showed'

  **Response**: Revised as suggested.

- L99: Do you mean forced QBO and SAO-like or forced QBO- and SAO-like or…?

  **Response**: The term "SAO and QBO like variability" was used by the authors of the paper referenced here (Pascoe et al., 2006), thus we have used the same terminology. In their paper they mean QBO- and SAO-like.

- L101: change to: ... we are here following suggestions that the upper atmosphere may be key to understand the…

  **Response**: Revised as suggested.

- L102: draw

  **Response**: Corrected

- L104: In the present study, we analyze the interactions of the …

  **Response**: Revised as suggested.

- L114: change 'vertical levels' to 'levels in the vertical'

  **Response**: Revised as suggested.

- L122: change 'were' to 'where'

  **Response**: Corrected.

- L124: remove one 'average'

  **Response**: Corrected.

- L133: Analogously

  **Response**: Corrected.

- L134: Rao et al. (2020)

  **Response**: Revised as suggested.

- L136: ...by contrasting the zonal winds to sonde radiosonde measurements from Singapore…

  **Response**: Revised as suggested:

- L154: MERRA-2

  **Response**: Corrected here and on one other occurrence.

- L154: deposition

  **Response**: Corrected across the manuscript.

- L160: Change '*' to '·'

  **Response**: The equations were removed following reviewer suggestion.

- L166: ...correspont to the …

  **Response**: Corrected.

- caption Fig. 1: ... at 2 hPa over ... and remove last sentence

**Response**: This figure was removed following reviewer suggestion.

- L183: In the austral winter of 2019, the QBO was in …

    **Response**: This section has been revised

- L189: 'comes to sit' ⇒ 'remains'

    **Response**: This section has been revised

- L198: form ⇒ from

    **Response**: This section has been revised

- L218: between 50 and 2 hPa

    **Response**: This section has been revised.

- L238 and L241: (Figure 8b)

    **Response**: This figure has been removed.

- L245: ... for these years.

    **Response**: Revised as suggested.

- L272: reversals

    **Response**: Corrected.

- L287: In 2002, the zonal mean….

    **Response**: Revised as suggested.

- L305 ...was it necessary…

    **Response**: Revised to "… it was necessary to constrain the model's global tropospheric winds and temperatures, and to further constrain the zonal wind in the equatorial atmosphere above 5 hPa to reanalysis fields."

- L321 heed?

    **Response**: Revised to "signal"

- L333-: Please move these links to the references or so.

    **Response**: This have been done.

**Responses to Reviewer #2**

Summary

Nordström and Seppälä (2021) used the MERRA-2 reanalysis to study the possible impact of the Quasi Biennial Oscillation and the Semiannual Oscillation on the SH SSWs in 2002 and 2019. They proposed that the interaction between QBO and SAO can improve the predictability of the SH SSW. However, such a link between the QBO-SAO combined interaction and the predictability of the SH SSW is not stable and not applicable to other year with similar QBO-SAO configuration in the upper stratosphere and mesosphere. Most of the results are fairly descriptive and different SH stratospheric responses to the QBO-SAO in different years are not well explained. Therefore, I would possibly recommend publication of the paper after a major revision is performed with the following comments considered.

**Major comments**

The introduction has a large bias toward the studies of QBO in literature. The authors should present a comprehensive review on the recent studies of the QBO in literature. Can the authors also give a complete review on the modelling of the QBO and possibly the SAO in literature? In addition, the QBO simulations from CMIP5/6 models have made a big progress recently, and the authors did not review in this paper (several JC, GRL, QJ, JGR papers report the simulation of the QBO in CMIP5/6 models, please refer to them for details). The Holton-Tan relation has also been assessed for the CMIP5/6 models in literature. I suggest the authors to read those new publications.

> **Response**: We have worked to improve the balance between discussion of the QBO and SAO in the introduction. Unfortunately we felt it was not possible to provide a long and detailed review, particularly as other reviewer comments strongly suggested shortening of the text in the introduction. As we only used reanalysis here, detailed discussion of the representation of the QBO in CMIP5/6 models did not seem necessary - again in the context of having been asked to shorten the introduction by another reviewer. We would be very happy to include more specific papers that are key to the QBO background for this work, but would kindly ask the reviewer to suggest which would be most appropriate.

L68-73: Factors influencing the occurrence of SSW in SH and NH have been reported in some papers, and summarized in Baldwin et al. (2021). Those factors include the QBO, ENSO, MJO, Solar cycle, and extratropical blockings. The authors did not read those original papers but see the summary in Baldwin et al. (2021). It is an efficient way to get an overall image on the latest study progress of SSW by reading the review paper Baldwin et al. (2021), but more details should be traced to the original paper. See the original paper (https://agupubs.onlinelibrary.wiley.com/doi/abs/10.1029/2019JD030826) Tables 1-3 for more details.

> **Response**: We have added more detail here as well as the suggested paper as a reference, and overall aimed to clarify the text relating to the factors as presented in the suggested paper.

The structure of the paper should be well trimmed and organized. The paper tries to emphasize the impact of the QBO and SAO, but the timeseries of the QBO and SAO are not shown in the paper. Show the QBO index at 10 hPa and the SAO index at 1 hPa at least. From the display of the paper, the maximum SAO westerlies are much stronger than the maximum SAO easterlies in some years. Readers do not know the SAO zonal wind asymmetry in other years. Further, the authors display some redundant figures (9-12). Is it necessary to composite the SSW group in different decades (or data streams in MERRA-2, L118-121)? Please condense the paper with your largest endeavor.

> **Response**: We have removed some of the redundant figures and grouped the reference years together as suggested.

> We also added the QBO and SAO time series at 10 hPa and 1 hPa (averaged over 5S-5N), respectively, as suggested. In addition, we included the latitudinal location of the zero wind line at the corresponding pressure levels to indicate the latitude where the winds turn from westerly to easterly. This figure, included below, shows how both the magnitude, horizontal extent, and timing display specific behaviour during 2002, 2017 and 2019 - these being the years when either SSWs or a disrupted antarctic winter polar vortex have been reported. We also highlight that the 1988 SSW reported by Kanzawa and Kawaguchi (1990) does not show this early winter behaviour. We find that the 1 hPa winds show speeds

of $< -10$ m/s in June while the zero wind line at 1 hPa simultaneously extends much further south than for other years, to latitudes between 30S-10S. At the same time the 10 hPa winds in June during these years are also easterly (with magnitudes between $-40$ to $-20$ m/s), with the zero wind line extending to 30S-15S. The 10 hPa wind speeds do not appear anomalous when contrasted to other years, but there is indication that the easterly winds do extend further south than for other eQBO years. This would suggest that the variability of the SAO (and its drivers) in particular plays a role here.

[Figure]

Top (left): Evolution of the equatorial (5S-5N) SAO zonal wind at 1 hPa level (m/s) from June to September. The grey lines show the wind for all eQBO years with the years 1988, 2002, 2017, and 2019 are highlighted with red, blue, orange, and yellow lines, respectively. Top (right) Latitudinal location of the zero zonal wind line at 1 hPa level. The small grey dot indicated the mean location while the large grey circle indicates the median location during the eQBO years. The grey bars show the $1 \pm \sigma$ deviation around the mean location. Years 1988, 2002, 2017, and 2019 are highlighted with red, blue, orange, and yellow markers, respectively. Bottom: As above, but for the zonal wind at the 10 hPa level for QBO.

Some papers might not be well understood by the authors. Gray et al. (2020) indeed tried to improve the prediction (rather than simulation or modelling, L304, L353) of the extreme polar vortex by considering the high-level impact.

Response: We apologise for unclear text here. We are aware that Gray et al. (2020) was a model study on how to improve forecasting of SSWs. We have revised the text to highlight that Gray et al. (2020) focus on prediction.

**Other comments**
L28: has occurred => have occurred

Response: Corrected

L33: The strong polar vortex in March 2020 and the Arctic ozone loss have also been reported (https://agupubs.onlinelibrary.wiley.com/doi/10.1029/2020JD033524).

Response: This paragraph was removed following a reviewer suggestion.

L44: The impact of SSW is wide, from polar region to midlatitudes. Please well read the reference and use the correct word.

**Response**: We revised this to:

*"The impacts of SSWs can influence the atmosphere from the polar region to mid-latitudes for months…"*

L49: The relationship between SSW and NAM in the NH is discussed for S2S model, and the uncertainty is also present (https://agupubs.onlinelibrary.wiley.com/doi/10.1029/2019JD031919).

**Response**: We have revised the text here and included the suggested reference.

L51: Here a reference should be added, like Allen et al. (2003) https://agupubs.onlinelibrary.wiley.com/doi/full/10.1029/2003GL017117

**Response**: We have added the Allen et al. (2003) reference as suggested.

L68-73, L81: Please see the discussion in Rao et al. (2020) for the possible impact of ENSO, QBO, MJO, Solar cycle on the SSW.

**Response**: Following on from the earlier comment, we have revised the text and include the suggested reference here.

L74: east and westward => eastward and westward

**Response**: Corrected

L124: average average => (delete one)

**Response**: Corrected

L135: is discussed => are …

**Response**: Corrected

L146: The author might use a composite analysis. Please write more specific.

**Response**: We have now moved to composite analysis of the eQBO years and clarify here that we checked all years individually but present the composite analysis as a reference point for the eQBO years.

L151: Only two SSWs are commonly reported for the SH in record. There was no SH SSW in 1988. (also see your L340)

**Response**: Years 1988 and 2017 have reported by Kwon et al. (2020) as having weak vortex events with onset date in August, similar to the year 2002 (Thompson et al., 2005).

The 1988 even has been reported as a SSW from Syowa station observations by Kanzawa and Kawaguchi (1990), but we have revised the text to emphasise that Kwon et al. (2020) and Thompson et al. (2005) suggest that this was a vortex weakening event.

Kwon, H., Choi, H., Kim, B.-M., Kim, S.-W., & Kim, S.-J. (2020). Recent weakening of the southern stratospheric polar vortex and its impact on the surface climate over Antarctica. Environmental Research Letters, 15(9), 094072. https://doi.org/10.1088/1748-9326/ab9d3d

Thompson, D. W. J., Baldwin, M. P., & Solomon, S. (2005). Stratosphere–Troposphere Coupling in the Southern Hemisphere. Journal of the Atmospheric Sciences, 62(3), 708–715. https://doi.org/10.1175/jas-3321.1

Kanzawa, H., & Kawaguchi, S. (1990). Large stratospheric sudden warming in Antarctic late winter and shallow ozone hole in 1988. Geophysical Research Letters, 17(1), 77–80. https://doi.org/10.1029/gl017i001p00077

L159, 161: there are two "theta" in the formulas, the authors should change the second one to "phi" (latitude). Please well read the book or reference and avoid those unnecessary errors.

**Response**: Yes the second one should have been $\phi$. The formulas were removed following another reviewer request.

L163-164: What mean did the authors use? Zonal mean? Or time mean? Please be more specific.

**Response**: This section has been removed following reviewer request.

L166: correspond => correspond to

**Response**: Revised

L167: What is the "upcoming EP flux"?

**Response**: The was meant to indicate the EP flux results shown in the following pages. We have revised the text to: " The EP flux results shown here were calculated from…"

L190: What latitude band do you describe?

**Response**: We have revised the figures and added additional ones to show this better and revised the text in this section to clarify the wording accordingly.

L198: form => from??

**Response**: Corrected.

L207-209: Could you provide the timeseries of the SAO from 1979 – nowadays?

**Response**: We have added a figure to address this comment, following the suggestion in the major comment above.

L210-225: This paragraph is too descriptive.

**Response**: With the revised figures we have also revised all text and added more quantitive discussion of the results.

L228-230: How regular is the SAO? See the major comment and provide the timesires of the SAO.

**Response**: We have added a new figure on the SAO variability following the major comment above.

L238: Figure 8b should be put in a pair of parentheses.

**Response**: With the revised figures we have also revised this part of the text.

L240: grammar error "is does"

**Response**: Corrected

L242-L277: This subsection is too descriptive and should be condensed.

**Response**: We revised the stream averaging to instead use a composite mean as a reference case. With this revision we also removed this section.

L317-319, L364: I don't think the SAO can help to explain why the SSW in SH is much less than the NH. This is mainly caused by the land-sea distribution responsible for the forced planetary waves.

**Response**: We propose that the SAO might contribute to this, we now include results showing that the SAO amplitude was particularly large in the years when SSW (or early weak vortex) events take place. This could help create conditions where the wave forcing is able to more effectively influence conditions in the SH. We discuss this aspect more in the manuscript as well as include new figures to quantify the magnitude of the SAO as well as EP flux divergence.

L320: It is nearly impossible to forecast an SSW with 2-3 months in advance.

**Response**: We do not claim this, but rather that we find some signals that in the SH cases appear 2-3 moths in advance. We have done additional analysis based on the reviewer comments that will hopefully clarify some of these points. We do not wish to claim that this would be the situation for the NH and state in the manuscript that NH cases need to be tested separately. We have revised the text here to clarify these points.

L326-327: The QBO is still a challenge for forecast systems. Not to mention the SAO. Please refer to https://journals.ametsoc.org/view/journals/clim/33/20/jcliD200024.xml

**Response**: We added the following text here with the recommended reference:

*"As noted by the multi-model study of Rao et al. (2020b), representation of the QBO is also remains a challenge."*

L330: read the original paper mentioned by Baldwin et al. (2021) for details.

**Response**: We added more detail into the introduction, including references, as suggested earlier and now simply state the following here:

*"As mentioned earlier, much work has been done in understanding both causes and implications of SSWs, particularly in the NH. Many interactions with large scale atmospheric modes or external forcing have been found to influence NH SSW occurrence, including the QBO, the ENSO, solar cycle, and the MJO. "*

L334-335: What do the authors mean "the MJO index was positive"? There are two MJO modes, and their combination presents a phase space. We use phases 1-8 to describe the MJO rather than "positive/negative".

**Response**: We have revised this to state that this is for the amplitude of the two leading models and does not contain information on the phase of the modes.

L344: into the SH => into the SH extratropics.

**Response**: Revised

L347-348: Incomplete sentence. It is a long phrase.

**Response**: We have revised this to

*"Changing zonal wind further influences wave propagation conditions, ultimately transferring the signal to the stratosphere and contributing to triggering of SSW conditions."*

Responses to Reviewer #3

The goal of this study is to show that zonal winds in the tropical stratosphere affected the development of the Southern Hemisphere minor sudden stratospheric warming (SSW) in 2019. The results are emphasized by comparisons of this winter with dynamical developments during two other disturbed SH winters and during quiet winters. This is an interesting topic and is timely since it investigates developments during the recent 2019 winter, which showed an enhanced level of disturbance that is rarely seen in the SH.

Overall, the paper lays out the authors' argument in a straightforward manner. However, the language is often lacking in precision so some steps along the way are over-simplified or not carefully described. The net effect comes across as a very qualitative description in which complexities are ignored or glossed over; at times the discussion is not well-grounded in dynamical theory. This could be remedied with an extensive revision focused on more careful description of the links between cause and effect, better definition of terms, and overall effort to avoid generalizations that do not hold up. It would also be very helpful if the figures were revised to better illustrate the points being made in the text. The comments below elaborate on these concerns. The aspects that are of most concern to this reviewer are those given in major comments 1, 4, and 6.

**Major comments**

1.  The paper is focused on the way that variations in tropical winds might contribute to the deceleration of the zonal winds in midlatitudes. The potential impact of the disturbed dynamics on the tropical wind is not discussed. The accepted understanding is that the stratopause SAO easterly winds in solstice periods develop in response to the global Brewer-Dobson circulation that is driven by planetary wave dissipation in the winter hemisphere. In other words, the anomalous EP flux divergence in particular southern winters could contribute to stronger SAO easterlies during those winters. That is not to say that the tropical winds cannot also affect the midlatitude dynamical disturbance but it is not appropriate to treat it as a one-way influence. The direction of influence is likely to be two-way; cause and effect cannot be easily separated.

    **Response**: The is a very important point and we have addressed this by adding further analysis to the manuscript, and also revising the wording in the text to highlight that two-way influence is likely. The revised figures for the years 2002 and 2019 show better that the anomalous EP flux convergence indeed likely contributes to the equatorial wind anomalies.

2.  Section 2.1: The paper by Kawatani et al (2020), which was cited in the manuscript, shows that the equatorial winds in the upper stratosphere in MERRA-2 have significant deficiencies. The winds in the lower mesosphere have even poorer agreement with observations. Some mention of these discrepancies is necessary.

    **Response**: We have addressed this by revising the text here to read:

    *"At 1 hPa MERRA-2 has been found to represent the easterly SAO in qualitative agreement with satellite derived winds (Kawatani et al., 2020). However, MERRA-2 has westerly bias compared to other reanalysis data and observations above 20 hPa. For the months considered here (June-September), Kawatani et al. (2020) show that the interannual variability in MERRA-2 SAO is comparable to other reanalysis data sets, suggesting that for our analysis, changes from year to year should be captured at a reasonable level."*

    Later, when presenting some of the new results we note the westerly bias again.

3.  It is often hard to see the features being discussed in the latitude x pressure figures (3, 4, 6, etc.). Can you make them wider in the latitude dimension by, for example, stacking two over two and/or cutting off the northern latitudes?

    **Response**: We have revised all figures and now only show latitudes 90S-10N in these types of panels.

4.  Besides being difficult to discern in the plots, the text describing the easterly winds is difficult to follow. One problem is labeling all easterly winds to be part of the QBO or SAO. For example (l. 218) "eQBO comes to lay between 50-2 hPa extending to about 40°S". The QBO is an oscillation in the tropics; the term is also sometimes applied to periodic signals elsewhere that could be affected by the tropical oscillation. What evidence do you have that this individual instances of easterly winds in mid-latitudes is part of the QBO rather than a response to other dynamical activity? It is good to keep in mind that the QBO and SAO are oscillations that are defined by a timeseries of winds at a given location. Take care when referring to winds with the same sign seen on a single occasion elsewhere in the atmosphere.

    **Response**: We have revised all the text in the result and now steer away from this type of description and only occasionally refer to SAO-like or QBO-like easterlies in this type of context.

5.  Figure 4 and discussion: The EP flux and its impact on the zonal wind are derived using zonal means and waves defined as perturbations from that mean. They cannot be applied to an arbitrary longitude sector. To look at the regional wave-mean flow process, you should apply analysis tools specifically developed for this purpose, such as the formulation given by Plumb (1985). I suggest to remove this figure and delete the discussion.

    **Response**: We have removed this figure as suggested and removed the corresponding discussion.

6.  Please define what you mean by "coupling of the SAO and QBO wind signals" used in reference to Figure 5. Do you mean to indicate that some dynamical interaction is "coupling" these winds? If so, please describe what it is. The terms "interact" and "interaction" are also used in the discussion of several other figures and in the discussion section but the nature of this interaction and the evidence that it is occurring are never presented.

    **Response**: By interaction we were referring to the merging of the two wind patters. We have revised the wording across the text to clarify this and aim to avoid the use of the word interaction.

7.  The speculation in the paragraph beginning at line 316 has several problems. First, it ignores the role that dynamics in the winter hemisphere have in driving the SAO and affecting its variability. Second, to support the speculation about the contrast with the NH, it is necessary to show or cite evidence that the development of the SAO is different there. This is not done.

    **Response**: We have revised this section to address these points. The text now reads:

    *"We propose that this early winter behaviour may aid in identifying conditions that lead to deceleration of the polar winds, which could then precondition the atmosphere for a SSW. For example in 2019, when there was enhanced upwards wave flux in August (Shen et al., 2020), the modulated waveguide in the stratosphere and above may have provided further optimal conditions for large disturbance to take place. It may also partially explain why SSWs are less common in the Southern Hemisphere: if the early and large SAO-QBO-like merging contributes to optimal conditions for SSW, not only is this dependent the QBO being in the correct phase, but also the SAO having a large amplitude during the early-mid winter. The SH winter typically experiences smaller amplitude SAO easterlies, while the NH winter experiences much larger*

*easterly winds (Smith et al., 2017). However, this hypothesis would need to be tested separately for the NH. It is also important to note again that the NH has higher winter planetary wave activity and variability than the SH.”*

**Minor comments**

The reason for analyzing the results from quiet years in streams is not clear. Do we learn anything from this that would not be equally apparent from treating all the quiet winters together? If so, please describe what it is that we learn and what about the different streams accounts for the differences. If not, it would be useful to eliminate this as an unnecessary complication.

**Response**: We have revised this and now treat all quiet years together as suggested.

(l. 36-37) “a reversal of the meridional temperature gradient, creating an easterly zonal wind” The meridional temperature gradient and vertical wind gradients are consistent. Be careful about saying that one is causing the other unless you have evidence that only one is being forced.

**Response**: We have removed this text following a suggestion by another reviewer.

(l. 46-47) “These clouds are where ozone depleting reactions occur (Solomon, 1999)” Not really. Reactions within clouds produce reactive species that then destroy ozone in the presence of sunlight.

**Response**: This text has been revised and we now only write:

*“…the warming of the stratosphere suppresses the formation of polar stratospheric clouds (Shen et al, 2020), which play a critical part in stratospheric ozone depletion (Solomon, 1999)”*

(l. 64-67) “The North Pole is ringed by mountain ranges, perfect for producing atmospheric waves (Duck et al., 2001). An enhancement of wave activity over winter causes disruption to the vortex, as the waves deposit their momentum at higher altitudes (Brasseur, 2005). However, Antarctica is enclosed by flat oceans, which don’t excite waves as effectively as mountains (Holton, 2012).” It is not necessarily the topography near the pole that is associated with planetary wave generation. It would be better just to say that the NH has higher winter planetary wave activity and variability than the SH.

**Response**: We have revised this section as suggested and it now reads:

*“NH has higher winter planetary wave activity and variability than the SH, thus leading to higher SSW occurrence in the NH.”*

(l. 86-87) “Westerlies maximise close to the equinoxes, whilst the easterlies maximise near the solstices (Brasseur, 2005)” Please be clear that this description applies for the SAO near the stratopause, not above.

**Response:** We have revised this to state

*“Westerlies near the stratopause…”*

(l. 138-139) “if the 10 hPa equatorial flow is easterly during the austral winter months of June/July” Do you mean if the monthly mean values are easterly for these two months or if the values are easterly for every day?

**Response**: We mean on average for the two months and now clarify this in the text. There are challenges due to period of the QBO, but we also now include a figure to show how the

10 hPa equatorial flow changes in the winter months for all the years taken as eQBO here. We also now address in the later discussion how this may affect the results.

In the section here we now write: *"For this study, eQBO is taken to be present if the mean June-July 10 hPa equatorial (5S-5N) zonal mean flow is easterly."*

(l. 156-157, l. 181, and elsewhere) "As planetary scale waves can only propagate where the zonal flow is westerly (eastward)" This should refer specifically to stationary waves. To be precise, insert "stationary" and also verify that the wave or waves that are driving the SSW under investigation are themselves quasi-stationary.
1 and eq. 2 are the quasi-geostrophic version of EP flux. The full values derived from the primitive equations would be preferable.

**Response**: The first text section was removed following reviewer feedback on this particular section.

The information on enhanced stationary planetary waves for the 2019 SSW has been added to the introduction, including references to other published works.

We now clearly specify in the text that *"Propagation of stationary waves requires westerly flow, thus the zero wind line forms a barrier for stationary wave propagation."*

(l. 189) "the zero wind line formed by the QBO and SAO subsides" What do you mean by subsides?

**Response**: This was not clear and was meant to indicate that it "moves down" and "descends". However, we have rewritten the results section in light of the additional results and the revised figures and this type of discussion was omitted, with more focus on quantitive description.

**Responses to Reviewer #4**

The authors propose a mechanism involving the interaction of the mesospheric semiannual oscillation (SAO) with the quasi-biennial oscillation (QBO) to explain the occurrence of SH weak vortex events. Overall I thought the authors' mechanism certainly warrants consideration and they provide some thoughtful, though qualitative, evidence for the important of this interaction. However, my main criticism is that there was little quantitative or statistical evidence provided to show that these processes are linked. Additionally, the authors make some comments (e.g. line 10-11, 339) that argue the SAO/QBO configuration is the primary ingredient for these somewhat rare SH events to occur, without really discussing the fact that tropospheric wave driving was extraordinarily large, for example, in August 2019 (Rao et al. 2020). For these reasons and the other major comments below, I suggest a major revision.

**Major Comments**

A general recommendation for the authors is to be careful or more exacting with their wording regarding this event- for example about calling it "the second SSW" (line 3, line 27-28, line 325). This event was not a major SSW (by technical criteria) like 2002, and if you mean to refer to minor events, there have been other minor SH events before too as mentioned on line 57 (though none as large as this one in terms of temperature). Maybe instead for line 3, you could state something like "while this was only the second time stratospheric temperatures have risen this rapidly in the Southern Hemisphere (SH) winter/spring". Also the number of 6 events per decade in the NH (or one almost every other year) is mentioned a few times but again that's referring to events that meet the major SSW criteria so it's not really comparable. The authors do try to clarify this point on line 55 but it would improve the text to be careful with wording throughout (or move the statement on line 55 to earlier to try to be clear from the beginning). I would disagree with the statement on line 61 as well, as 2019 saw no wind reversal at 60S and 10 mb; additionally, there is evidence to suggest that dynamics were quite different between the 2002 and 2019 warmings, so I'd rephrase.

> **Response**: We have clarified the text throughout the manuscript on this regard and now clarify that the 2002 case is the only known major SSW in the SH (by the WHO criteria), while the other cases discussed are either minor SSW and/or weak vortex events as previously shown by Thompson et al. (2005) and Kwon et al. (2020).
>
> Kwon, H., Choi, H., Kim, B.-M., Kim, S.-W., & Kim, S.-J. (2020). Recent weakening of the southern stratospheric polar vortex and its impact on the surface climate over Antarctica. Environmental Research Letters, 15(9), 094072. https://doi.org/10.1088/1748-9326/ab9d3d
>
> Thompson, D. W. J., Baldwin, M. P., & Solomon, S. (2005). Stratosphere–Troposphere Coupling in the Southern Hemisphere. Journal of the Atmospheric Sciences, 62(3), 708–715. https://doi.org/10.1175/jas-3321.1

There needs to be some more analysis to connect the changes in the critical line/wave propagation in the subtropics with the momentum changes happening in the polar regions. The authors have a couple nice figures (Figure 3,4, and 6) qualitatively showing a potential connection. But what's not clear is if the momentum changes over the polar region are just happening simultaneously as the changes in the critical line, or if these things are truly related. Perhaps there is some way to examine the correlation between the momentum flux convergence near 60S/1 hPa with some measure of the critical line at 1 hPa (latitudinal location for example), for all years in the record during July, and see if 2019 and 2002 correspond to exceptional conditions on that scatterplot. This might also provide better quantification for how other years are different (e.g. Figure 9). Similarly, in section 3.3.2 there is a lot of qualitative discussion of the SAO-QBO interaction (e.g. lines 243, 250) but it would greatly improve the arguments if this relationship and its link to the polar vortex could be better quantified. For example, is the latitude of the SAO-QBO interaction the most important feature? The vertical extent? The magnitude of the SAO/QBO winds? These factors are brought up on lines 260-261 and 340 but can these be quantified in a more robust statistical manner?

> **Response**: We have added new analysis to contrast the EP flux divergence (in terms of the effect on zonal wind acceleration/deceleration) and the location of the zero wind line at 1 hPa level, using scatter plots as suggested.

Using revised version of the zonal average figures (with restricted latitudinal range) we determined representative locations for upper stratospheric and lower stratospheric EP flux responses and contrasted the upper stratospheric (1-10 hPa) EP flux divergence at 50-40S with the latitude of the zero wind line at 1hPa, and similarly for lower stratosphere (50-70 hPa) at 60-40S. This was done for two average periods for the zero wind location (mid-June to mid-July and late-June to late-July) and a range of times for the EP flux divergence to capture mid-winter and later winter behaviour. The result is shown below, with the weak vortex years indicated with red colour.

[Figure]

Latitude of the zero wind line (x-axis) at 1 hPa level (left panels: 15 June - 12 July; right panels: 29 June - 26 July) versus EP flux divergence (divEP, m/s/day, y-axis).
a-b) Mean EP flux divergence at 50S-40S and between 1-10 hPa, times as given in titles; c-f) Mean EP flux divergence at 60S-40S and between 50-70 hPa, times as given in titles.

Looking at panel a) for the earlier time period (15 June-12 July), both years 2019 and 2002 show EP flux divergence acting to decelerate the zonal flow at the rate of 2-3 m/s/day in the upper stratosphere. Similar behaviour is seen for the year 2017 in panel b) (the later time period) with zonal flow deceleration of about 4.5 m/s/day. All these years have a zero wind line that extends further South than during most years. The weak vortex year 1988 does not appear anomalous in these scatter plots. In the lower stratosphere, the weak vortex years (except for 1988) all again present higher zonal flow deceleration rates in the Earlier time period. We also see that years 1980 and 1990 have zero wind lines that extend to 10S-15S, but these years do now exhibit similar EP flux divergence levels as the weak vortex years.

We added further time periods to check for the presence of the previously reported anomalous tropospheric forcing. Panels e)-f) show the results for EP flux divergence in the lower stratosphere

averaged over 10 August-6 September, around the time Shen et al. (2020) have reported anomalous tropospheric wave forcing entering the stratosphere. We contrast this to the location of the zero wind line in the two earlier time periods and find the situation to be very similar to that in panel a), corresponding to EP flux divergence in 15 June-12 July.

[Figure]

Top (left): Evolution of the equatorial (5S-5N) SAO zonal wind at 1 hPa level (m/s) from June to September. The grey lines show the wind for all eQBO years, with the years 1980, 1988, 1990, 2002, 2017, and 2019 highlighted with coloured lines. Top (right) Latitudinal location of the zero zonal wind line at 1 hPa level. The small grey dot indicates the mean location while the large grey circle indicates the median location during the eQBO years. The grey bars show the $1 \times \sigma$ deviation around the mean location. Bottom: As above, but for the zonal wind at the 10 hPa level for QBO.

When looking at the equatorial zonal wind at 1hPa and 10hPa (and the associated zero wind line latitudinal locations, see figure below) we find that the common features for the years 2002, 2017 and 2019 are:

1.  Large easterly equatorial flow at 1 hPa (10-20 m/s) in June, and anomalous (beyond 1 $\sigma$ from the mean latitude) Southward extension of the zero wind line at 1 hPa from mid-June.

2.  Large easterly equatorial flow at 10 hPa (20-35 m/s) established and sustained from June. These years show a Southward extension of the zero wind line at 10 hPa, but not necessarily beyond 1 $\sigma$ from the mean latitude.

Some but not all of these conditions are present for 1980, 1990 and 1988, suggesting that both equatorial wind patterns contribute to the weak vortex events.

We have included these new figures in the revised manuscript and revised all of the text to reflect these results, also adding much more quantification based on the results presented in these figures. Note that, for clarity and consistency, for the second of the figures to include in the manuscript we would highlight only the earlier identified weak vortex years instead of all six years highlighted in this version with the two additional years described in the text.

Shen, X., Wang, L., & Osprey, S. (2020). Tropospheric forcing of the 2019 Antarctic sudden stratospheric warming. Geophysical Research Letters, 47, e2020GL089343. https://doi.org/ 10.1029/2020GL089343

It wasn't clear to me why it was necessary to break the averaging into MERRA-2 streams. Is there justification for doing this? Maybe there are large jumps introduced (e.g., Long et al. 2017), but I wouldn't

think it would have such a substantial effect on weekly averages. This made for a lot of unnecessary figures in my opinion, and it is not clear that a lot of the differences between, e.g. Figures 9, 10, and 11, aren't just due to sampling (only a few cases in each composite)- thus this was not particularly meaningful for me to be able to draw any conclusions. Why not just composite all the non-weak vortex years together? Or just pick a couple of years of interest and just show those individual years? (the averaging could be introducing differences just from differences in the timing of SAO-QBO interaction between years, for example). Similarly, the statements on lines 257-260 and lines 281-285 are implying some decadal differences based on arbitrary averaging over streams, but I'm not sure these statements are justified (what would be a physical explanation for why these relationships would be different over time?).

> **Response**: We have revised this to move away from the stream averaging and now use a full non-weak vortex composite mean instead as suggested. With this we also now show the 7-day means throughout the months our analysis focused on.

The authors make an interesting case for the role of mesosphere-stratosphere interactions, but I think they ignore too much the role of tropospheric forcing and/or internal stratospheric processes. For example I think Figure 11 shows cases in which the SAO-QBO interaction is very similar to 2019 and 2002 yet no SSWs happened- why? Particularly during 2019 when tropospheric wave forcing was at record high levels, I think this is a mistake to not mention this or explore the combined effect of these two features- perhaps the mesospheric wind interaction is a necessary but not sufficient condition for a SSW to occur (or likewise for the tropospheric wave driving; perhaps you need both factors). For this reason I think statements such as 338-339 are overstated, and should be reworded unless stronger evidence is provided that this interaction can actually "trigger" the event.

> **Response**: This is a very important point and we have carefully revise the wording around the importance of different drivers. We have added more discussion on the enhanced tropospheric wave forcing (Shen et al. 2020), also noting that this has been reported start in August. We now also present extended time period in our figures, showing the enhanced upwards EP flux from the troposphere happening consistently with the timing shown by Shen et al. (2020), as well as having added to the above described scatter analysis, a time period covering when the enhanced tropospheric wave forcing times.
>
> Shen, X., Wang, L., & Osprey, S. (2020). Tropospheric forcing of the 2019 Antarctic sudden stratospheric warming. Geophysical Research Letters, 47, e2020GL089343. https://doi.org/ 10.1029/2020GL089343

**Specific comments**
Line 2, also line 30: specify the level/latitudes that this wind reversal and temperature rise occurred at (or at least something general like "polar cap" and "mid-stratosphere").

> **Response**: We have now added this information as suggested.

Line 4, line 62: This is one important aspect that is thought to drive SSWs but one difficulty is that the only known SSW in the SH in 2002 was likely not due to tropospheric wave driving but resonant amplification internal to the stratosphere (see Albers and Birner 2014, Esler papers). A number of NH events are also thought to be caused by this, with the link to tropospheric wave forcing not always prevalent (de la Camara et al. 2019). So either also mention this mechanism or caveat these statements so that it's clear this is not the only possible mechanism.

> **Response**: We revised the abstract text to:
>
> *"Amplification of atmospheric waves during winter is thought to be one of the possible trigger SSWs, although other mechanisms are also possible."*
>
> We now also cite the Albers and Birner 2014 and de la Camara et al. 2019 papers in the results and introduction sections, respectively.

Line 33-35: This is true but the other key dynamical difference between a final warming and a sudden warming is that for the SSW the vortex recovers back to westerlies afterwards.

**Response**: This paragraph was removed following a suggestion from another reviewer.

Line 31, 34: Here, and throughout, there are a lot of textbook references for things that are more or less "facts"- I don't think every statement that is considered general knowledge needs a reference (or at least, there might be more appropriate references to cite; for example on line 34 & 39 I suggest instead the review on SSWs, Baldwin et al. 2020 in Reviews of Geophysics).

**Response**: We have removed the textbook references as suggested and use e.g. the Baldwin et al. review instead.

Line 47-49: Rephrase; they don't have to be easterlies to descend to the troposphere; stratosphere-troposphere coupling occurs when the winds are anomalously strong, e.g. positive NAM, as well.

**Response**: This has been revised to read:

*"The anomalous winds from SSWs can also influence stratosphere-troposphere coupling, impacting the Southern and Northern Annular Modes (SAM and NAM) (Taguchi and Hartmann, 2005; Shen et al., 2020; Baldwin et al., 2021)."*

Line 54, and line 160: From Figure 2d, I'd argue that the vortex is not centered over the pole once the SSW occurs (even at 40 hPa it's fairly displaced).

**Response**: This section has been revised to not include this description and we have left the figures out based on reviewer feedback.

Line 63: it's not really the North pole but the middle latitudes where the mountain ranges matter for wave forcing.

**Response**: From reviewer feedback, this has been revised to "*NH has higher winter planetary wave activity and variability than the SH, thus leading to higher SSW occurrence in the NH.*"

Line 76, and line 101: easterly QBO at what level? Note that a lot of the Holton-Tan relationships are looking at QBO at 40-50 hPa, not 10 hPa as done in this study.

**Response**: We now specify the QBO pressure level ranges in this paragraph.

Line 82: westerlies maximise at what level?

**Response**: We now specify that this is for the stratopause level (1 hPa).

Line 140: It's not clear if "all experienced mesospheric wind reversals in October" is meant to refer to 1988 and 2017 or to the other years identified- also why October, isn't this when the final warming or seasonal transition begins to occur? Seems late compared to 2002 and 2019.

**Response**: We have revised text in this section to:

*"To contrast the SSW years of 2019 and 2002 to others with similar large scale equatorial flow conditions, other years with equivalent, i.e. eQBO phase, conditions during the austral winter months were analysed as a reference. Additionally, for the reference dataset, we leave out the early onset date (August onset) weak vortex years of 1988 and 2017 (Kwon et al., 2020)."*…

…*"As noted above, the years 1988 and 2017 were considered separately from the reference data set, due to the early vortex weakening events. "*

Lines 143-145: For a list of years when the SH polar vortex was weak, see Lim et al. 2019 (https://www.nature.com/articles/s41561-019-0456-x)

**Response**: We have revised the manuscript using the table provided by Kwon, H., Choi, H., Kim, B.-M., Kim, S.-W., & Kim, S.-J. (2020). Recent weakening of the southern stratospheric polar vortex and its impact on the surface climate over Antarctica. Environmental Research Letters, 15(9), 094072. https://doi.org/10.1088/1748-9326/ab9d3d focusing on early season vortex weakening events.

Line 168-169: "as may happen with wQBO"- can you clarify what is meant here?

**Response**: The westerly phase of the SAO is known to sometimes initiate (or appear to initiate) a westerly QBO phase. As we have removed the figures that were discussed here, this text has also now been removed.

Figure 5, figure 7: 15-20S is a narrow latitude band- which makes it difficult to see the easterly SAO winds in the tropics. What about 0-30S or 0-20S instead? Perhaps that would make the descent between the SAO and the QBO also clearer. The "downward connection" between the SAO and QBO in these figures was not particularly compelling for that reason; particularly for 2002, there only seemed to be some interaction for a week in mid-June, when the SSW didn't happen until late September (the connection is easier to see for this reason in Figure 6).

**Response**: We have revised all figures and since the SAO and QBO wind pattern evolution can now be seen more clearly in other new figures, these two figures did not add much so we decided to remove them entirely.

Line 225, 276: Figure 8d- there's not a wind reversal near 60S except above 1 hPa; and isn't this just the final warming/season transition? It's not clear to me the utility of showing late October for this reason since it seems likely there are different processes at play (radiative changes driving the seasonal cycle). Why not show what Sept looked like instead (as in Figure 3 and 6) to show what was different about 2017 so that it didn't result in a warming, given otherwise strong similarities with the critical line/QBO/SAO in June?

**Response**: We have revised all figures and now present all time periods consistently so the different years can be contrasted more easily.

Line 257: smaller in what way? Can this be quantified? (see major comments)

**Response**: We have revised this entire section and now include more quantitive measures through the text.

Line 265: I don't understand why there are parentheses around polar and low-latitude, but I'm also not sure this sentence is clear.

**Response**: We have revised this to

*"Change in the latitudinal location where winds shift from westerly to easterly influences the waveguide in the upper stratosphere-lower mesosphere, resulting in easterly momentum being deposited on the equatorward side of the polar vortex from early winter."*

Line 283-4: "We didn't find evidence of easterly momentum being deposited throughout the winter as we did for 2002 and 2019"; this is a bit circular, but it also shows the issue with causality in this study. The other years didn't have build up of easterly momentum and there was no SSW; but is this a reflection of differences in the SAO-QBO relationship or the fact that there was no SSW due to other processes? This lack of causality is reflected in the statement on line 286, where you mention the years you did have an SAO-QBO interaction but there was no easterly momentum build up or wind reversal.

**Response**: We have removed this and the previous paragraph and now here write in context of the added new analysis:

*We found that the weak vortex years of 1988 and 2017 show a similar SAO-QBO wind pattern like merging in July. However, neither of these years show a poleward shift in the zero wind line location as early as was seen for 2002 and 2019. Causes of these differences could be investigated further in a*

*detailed study. We note that in 2017, the changes in dynamics were enough to stifle the growth of the ozone hole (Klekociuk et al., 2020). Our analysis of all other years with similar background QBO conditions in MERRA-2 did not reveal similar behaviour with early winter sustained momentum deposition and similar SAO-QBO like easterly wind interaction in early winter.*

Minor discussion comment/line 303-304: Some attempts have been made to look at the upper stratosphere in the SH winter to make predictions for the following spring which should be acknowledged here:
Lim, E.-P., Hendon, H.H. & Thompson, D.W.J., 2018. Seasonal Evolution of Stratosphere-Troposphere Coupling in the Southern Hemisphere and Implications for the Predictability of Surface Climate. Journal of Geophysical Research: Atmospheres, doi: 10.1029/2018JD029321.
Byrne, N.J. & Shepherd, T.G., 2018. Seasonal persistence of circulation anomalies in the Southern Hemisphere stratosphere, and its implications for the troposphere. Journal of Climate, doi: 10.1175/JCLI-D-17-0557.1.

> **Response**: Thank you for pointing these out, we have added both references to the discussion section.

Line 305, 344: 20-30 days is not typical for SSW predictability (maybe a handful of events in the record have been predictable on timescales that long). Refer to Domeisen et al. 2020 (part I)- not the one that is cited in the next sentence- typical predictability limits are 10-15 days in the NH. However, it might turn out that SH SSWs are predictable at longer leads, if they are more influenced by the mesosphere/stratosphere interaction than the NH.

> **Response**: We have revised this section to clearly state that the typical predictability is 10-15 days in the NH and referenced Domeisen et al. 2020 (part I).

Line 316: One question I had was why was the focus of this study on the SH rather than the NH (which would greatly increase your sample size?). Is this something presented already in the Gray et al. (2020) paper? Perhaps it should be stated somewhere why the focus here is on the SH (maybe it is and I missed it). In terms of the QBO, ENSO, and MJO influence it should be mentioned that the Rao et al. 2020 paper considers several of these factors on the 2019 SH warming.

> **Response**: In deed we were motivated by the many unknowns when it comes to SH SSWs. Most of the research on SSW has focused on the NH, and extending this analysis to the NH would be the next logical step. However, this study was primarily motivated by the severe impacts of the 2019 SH SSW experienced in Australia. The strong winds carried the Australian bushfire smoke into New Zealand and as New Zealand based scientist, we were motivated to investigate any possible precursors to the SSW taking pace earlier in the winter, improving lead timed for predictions in our regions of the world.

Line 320: what is meant by the "MJO index was positive"? (the amplitudes of the EOF1/2 in the MJO phase space were positive?) What was the phase?

> **Response**: We have revised this to state that this is for the amplitude of the two leading models and does not contain information on the phase of the modes.

Line 348: suggest non-gendered associations such as "their boreal counterpart".

> **Response**: We have removed this.

Line 347-348: Are their anti-symmetrical aspects of the QBO/SAO across hemispheres that could explain a difference between NH and SH frequency of SSWs? To me there seems an obvious difference in tropospheric wave driving between the two, but if there are clear differences in the upper atmosphere that could explain frequency differences it would be interesting to describe or speculate on here.

> **Response**: The easterly winds of the upper stratospheric SAO have been reported to be stronger in December, during the Northern Hemisphere winter, while the "second cycle", which maximises in June is reported to typically be weaker. We include new figures that show that the years of the SH weak vortex years are consistent with times with higher amplitudes of the SAO easterly winds in June, which

would point towards the anti-symmetric aspect between the hemispheres. We now discuss this aspect more in the paper.

**Technical corrections**
Line 22: "Annual" should be "Annular"
Line 42: should be "forming a 'comma shape'"
Line 46: replace "intrusion" (which has a very specific meaning tied to stratosphere-troposphere exchange) with "mixing"
Line 50: correct misspelling of "split"
Line 57: remove "of"
Line 63: replace "geography" with "topography"or "orography" perhaps
Line 85: change to "roughly in June"
Line 117: change "were" to "where"
Line 139: change "left out of these groups" to "considered separately"
Line 146, 170, 175, 294, 329: change "deposit" to "deposition"
Line 182: change "form" to "from"
Line 187: change to "western side of Antarctica"
Line 192: change to "before it connects"
Line 223: not sure what is meant by "subsides" in this context- "moves down"?
Line 253: change "on" to "of"
Line 261: change "there" to "their"
Line 321: change "difference" to "differences"
Line 331: change "influencing" to "influences"

**Response**: All technical corrections have been implemented as suggested.

---

## Referee Report (RR1)

Second review of "Does the coupling of the semiannual oscillation with the quasi-biennial oscillation provide predictability of Antarctic sudden stratospheric warmings?" by Nordström and Seppälä

The authors have been very responsive to the comments of the reviewers and have thoroughly revised their manuscript. It is now much improved and, it seems to me, does a good job of getting across a message to the reader.

I have some minor comments which mainly have to do with the presentation.

Minor comments

I appreciate that the authors are trying to pull together their findings in graphical form in Figure 7. However, I found that it took some real effort, going back and forth between the text and the figure, to actually absorb this information. A good goal would be to have one figure that captures the message that you are presenting in a way that is accessible. One possibility would be to extract the four panels in the top two rows into a separate figure since they at least do not have the confusion of different dates used for the different axes. Then the panels from the bottom row (or maybe only one of them is necessary) could be prominently labeled to indicate that you are comparing early-season zero wind line with late-season divEP.

(l. 316) "The early winter equatorial SAO-QBO wind pattern interaction with the simultaneous EP flux convergence, and subsequent modulation of the waveguide, reflects mid-latitude waves up and pole-ward, resulting in deceleration of the equatorward side of the polar vortex above 100 hPa." Do you actually mean wave reflection? In principle a wave guide can affect the direction of wave propagation in a way that is not a reflection but is a guide, as the term suggests.

(l. 354-355) It is OK to mention the MJO amplitude but seems a bit of a stretch (even with the qualifier "could") to imply that it plays a role in these events. You do not show anything about this index in quiet years or how it affects planetary wave generation and propagation. I suggest remove the last sentence of this paragraph.

Editorial comments:

1.  (l. 24) "the easterlies around 60°S at 10 hPa reached ~60 ms$^{-1}$" Do you mean only at some longitudes? It's confusing because the first part of the sentence describes zonal mean zonal winds.
2.  (l. 36-37) "shrunk the ozone hole to its smallest size ever observed". Maybe better to say something like "shrunk the ozone hole to its smallest size since its onset in 1980".
3.  (l. 58-59) "It is widely thought that SSWs are the product of an interaction between planetary waves and the atmospheric mean flow (Matsuno, 1971)." The importance of wave-mean flow interaction in SSW has been demonstrated in hundreds of studies. The term "thought" comes across as dismissive of this well-established understanding. It would be better to say something like "Numerous investigations have demonstrated that …"
4.  (l. 87) remove "generally understood to be" (i.e., it is stronger)

5. (l. 300) "zonal wind reversal is less focused" What does this mean?